# Defective ribosomal products challenge nuclear function by impairing nuclear condensate dynamics and immobilizing ubiquitin

Laura Mediani[1], Jordina Guillén-Boixet[2], Jonathan Vinet[1,3], Titus M Franzmann[2], Ilaria Bigi[1], Daniel Mateju[2], Arianna D Carrà[1], Federica F Morelli[1], Tatiana Tiago[1], Ina Poser[2], Simon Alberti[2,4] & Serena Carra[1,*]

## Abstract

**Nuclear protein aggregation has been linked to genome instability and disease. The main source of aggregation-prone proteins in cells is defective ribosomal products (DRiPs), which are generated by translating ribosomes in the cytoplasm. Here, we report that DRiPs rapidly diffuse into the nucleus and accumulate in nucleoli and PML bodies, two membraneless organelles formed by liquid–liquid phase separation. We show that nucleoli and PML bodies act as dynamic overflow compartments that recruit protein quality control factors and store DRiPs for later clearance. Whereas nucleoli serve as constitutive overflow compartments, PML bodies are stress-inducible overflow compartments for DRiPs. If DRiPs are not properly cleared by chaperones and proteasomes due to proteostasis impairment, nucleoli undergo amyloidogenesis and PML bodies solidify. Solid PML bodies immobilize 20S proteasomes and limit the recycling of free ubiquitin. Ubiquitin depletion, in turn, compromises the formation of DNA repair compartments at fragile chromosomal sites, ultimately threatening cell survival.**

**Keywords** defective ribosomal products; membraneless organelles; molecular chaperones; nucleus; proteostasis
**Subject Categories** DNA Replication, Repair & Recombination; Post-translational Modifications, Proteolysis & Proteomics; Protein Biosynthesis & Quality Control
**The EMBO Journal (2019) 38: e101341**

## Introduction

The fidelity in the flow of genetic information from DNA to RNA and, finally, to protein is essential for cellular life. However, many errors can occur during DNA replication, transcription, and translation, which lead to the production of mutated or truncated polypeptides that cannot acquire their native folding state, impairing protein homeostasis (Kapur & Ackerman, 2018). Imbalances in protein homeostasis, in turn, lead to cell dysfunction, aging, and disease (Klaips *et al*, 2018). The main sources of misfolded proteins in mammalian cells are newly synthesized polypeptides that have not yet reached their native state and defective ribosomal products (DRiPs), which result from the misincorporation of amino acids, premature translation termination, damaged mRNAs, or DNA mutations (Schubert *et al*, 2000). DRiPs also include small translation products that are generated outside of protein-coding regions by so-called pervasive translation. Indeed, ribosome occupancy has been observed for many 5′ untranslated regions (5′-UTRs) and could potentially generate substantial amounts of aberrant peptides even under physiological conditions (Ingolia *et al*, 2014). These peptides are thought to be highly unstable, but their biological effects remain to be explored.

Regardless of their origin, once DRiPs are synthesized, they are recognized by specific protein quality control (PQC) machineries. These include the chaperones of the HSP70 family, which co-translationally bind to nascent polypeptides (Hartl & Hayer-Hartl, 2002), and VCP. Both HSP70s and VCP bind to DRiPs, assist their ubiquitination, and promote their degradation by the proteasome (Qian *et al*, 2006; Verma *et al*, 2013). Surprisingly, although DRiPs represent the major source of misfolded proteins in cells, how DRiPs are handled and cleared, and the consequences of mishandling them are largely unknown. We recently showed that DRiPs can accumulate inside membraneless compartments called stress granules (SGs) and promote their conversion from a liquid-like dynamic state into a solid-like aggregated state, with important biological and pathological implications (Ganassi *et al*, 2016; Mateju *et al*, 2017). However, how mammalian cells cope with DRiPs is still largely unresolved.

Here, we report that DRiPs rapidly diffuse into the nucleus, where they compromise nuclear protein homeostasis (or

1  Centre for Neuroscience and Nanotechnology, Department of Biomedical, Metabolic and Neural Sciences, University of Modena and Reggio Emilia, Modena, Italy
2  Max Planck Institute of Molecular Cell Biology and Genetics, Dresden, Germany
3  Genomic and post-Genomic Center, IRCCS Mondino Foundation, Pavia, Italy
4  Technische Universität Dresden, Center for Molecular and Cellular Bioengineering (CMCB), Biotechnology Center (BIOTEC), Dresden, Germany
   *Corresponding author. Tel: +39 0592 055265; E-mail: serena.carra@unimore.it
   [The copyright line for this article was changed on 1 August 2019 after original online publication.]

proteostasis). Maintenance of the nuclear proteome is important to preserve genome integrity. Indeed, many age-related neurodegenerative diseases, which are characterized by the presence of nuclear protein aggregates and proteostasis impairment, are also linked to genome instability (Shibata & Morimoto, 2014; Klaips *et al*, 2018). Although nuclear protein aggregation induces mutagenesis and DNA damage (Shor *et al*, 2013; Shibata & Morimoto, 2014), we do not understand how these processes are linked at the molecular level. Most importantly, we only have a very rudimentary understanding of how the nucleus copes with nuclear protein misfolding.

We find that, once in the nucleus, DRiPs accumulate in nucleoli and PML bodies, which are membraneless organelles similar to cytoplasmic SGs (Ganassi *et al*, 2016). If not properly cleared with the assistance of chaperones, DRiPs promote nucleolar amyloidogenesis and convert PML bodies from a dynamic state into a solid state. These solid PML bodies sequester large amounts of ubiquitin and 20S proteasomes. Immobilization of proteasomes and ubiquitin at solid PML bodies, in turn, compromises the formation of ubiquitin-dependent DNA repair compartments at fragile chromosomal sites, with deleterious consequences for cell fitness. Altogether, these data provide a mechanistic explanation of how nuclear proteostasis imbalance can compromise genome integrity.

# Results

### DRiPs accumulate in the nucleolus and promote the formation of amyloid bodies

Maintenance of the nuclear proteome requires transport of proteins through nuclear pores. This transport process is highly selective, although proteins with low molecular weight (< 40 kDa) can also passively diffuse through these pores (Lusk & King, 2017). DRiPs include prematurely terminated polypeptides with low molecular weight, which could passively enter into the nucleus if not properly recognized and degraded by the cytoplasmic PQC machineries. Whether DRiPs can translocate into the nucleus of mammalian cells is largely unknown. Here, we addressed this question by following the subcellular distribution and fate of nascent chains labeled with an analog of puromycin (OP-puro). Surprisingly, DRiPs could be detected inside the nucleus, where they accumulated in microscopic large clusters that resembled nucleoli. Using HeLa cells stably expressing GFP-tagged nucleolin (GFP-NCL) (Poser *et al*, 2008), a marker of nucleoli, we confirmed that a large fraction of DRiPs selectively accumulated in nucleoli of growing, unstressed cells. DRiPs were already detected in nucleoli after 15 min of exposure to OP-puro and their accumulation increased over the time-course of 2 h (Fig 1A and B, and data not shown). As control, co-treatment of HeLa cells with OP-puro and the translation inhibitors cycloheximide (CHX) or anisomycin (ANS) abrogated DRiP synthesis and accumulation inside nucleoli (Fig EV1A). During the recovery time, DRiPs were cleared from nucleoli; by contrast, when the proteasome was inactivated with the inhibitor MG132, DRiPs remained inside nucleoli for much longer (Fig 1A and B). This demonstrates that accumulation of DRiPs within nucleoli is reversible and suggests that removal of DRiPs from nucleoli requires the proteasome.

We further studied the distribution and size of DRiPs by subcellular fractionation, which confirmed the enrichment of puromycylated proteins with low molecular weight in the nucleoplasm and nucleoli of puromycin-treated cells (Fig 1C). Then, we asked whether DRiPs are actively imported inside the nucleus or, instead, passively diffuse through the nuclear pores because of their low molecular weight (Fig 1C). To differentiate between these two possibilities, we used ivermectin, a well-known inhibitor of nuclear import (Wagstaff *et al*, 2012). As positive control for ivermectin treatment, we verified the subcellular distribution of GFP-TDP43, which is mainly located inside the nucleus of untreated cells. As expected, upon treatment with ivermectin, a large fraction of GFP-TDP43 was retained in the cytoplasm (Fig EV1B). However, DRiPs were enriched inside nucleoli irrespective of active nuclear import (Fig EV1C). We conclude that a fraction of low molecular weight DRiPs is not targeted by the cytoplasmic PQC, diffuses into the nucleus, and accumulates in nucleoli.

Is nucleolar function affected by the accumulation of DRiPs? To address this question, we first investigated nucleolar function by measuring the levels of precursor and mature ribosomal RNAs (rRNAs). As positive control, cells were treated for 4 h with actinomycin D (Act. D), an RNA polymerase inhibitor that potently suppressed the synthesis of precursor rRNAs (Fig EV1D: 45S, 18S 5′J, and 5.8S 5′J). By contrast, 4 h of treatment with puromycin alone or combined with MG132 only caused a mild increase of precursor rRNAs 45S and 18S 5′J, as well as mature 18S (Fig EV1D). These results indicate that DRiP accumulation in nucleoli slightly affects rRNA 18S processing, without having a major impact on mature rRNA levels. Next, we determined the mobility of nucleolar proteins in the presence and absence of DRiPs by fluorescence recovery after photobleaching (FRAP) experiments. We used HeLa cells stably expressing GFP-tagged nucleophosmin 1 (GFP-NPM1), a marker of the granular compartment, and nucleolin (GFP-NCL), a marker of the dense fibrillar compartment (Poser *et al*, 2008). Remarkably, the dynamics of NPM1 and NCL in the presence or absence of DRiPs were very similar; moreover, we found no evidence for altered mobility of NPM1 or NCL upon concomitant treatment with OP-puro and the proteasome inhibitor MG132, which led to an even greater accumulation of DRiPs within nucleoli (Fig EV1E and F). Altogether, these findings suggest that DRiPs may be sequestered into specific nucleolar subcompartments, preserving nucleolar dynamics. In agreement with this idea, previous reports described the accumulation of various proteins in nucleolar subcompartments of cells exposed to different stressors, such as heat shock, acidosis, or prolonged treatment with MG132. These nucleolar aggregates were referred to as amyloid body (A-body) or the nucleolar aggresome (NoA), respectively (Latonen *et al*, 2011; Jacob *et al*, 2013; Audas *et al*, 2016).

Thus, we next tested whether DRiPs contribute to nucleolar amyloidogenesis under different stress conditions. First, we studied the link between DRiP accumulation and nucleolar amyloidogenesis under heat shock. Heat shock-induced A-bodies can be visualized with the dye Amylo-Glo (Fig 2A; Audas *et al*, 2016). We found that the translation inhibitor cycloheximide abrogated the accumulation of DRiPs in nucleolar subcompartments (Fig 2B) and it also prevented the formation of A-bodies in heat-shocked cells (Fig 2A). By contrast, inhibition of transcription with Act. D (Fig EV2A) did not prevent the accumulation of DRiPs in nucleolar

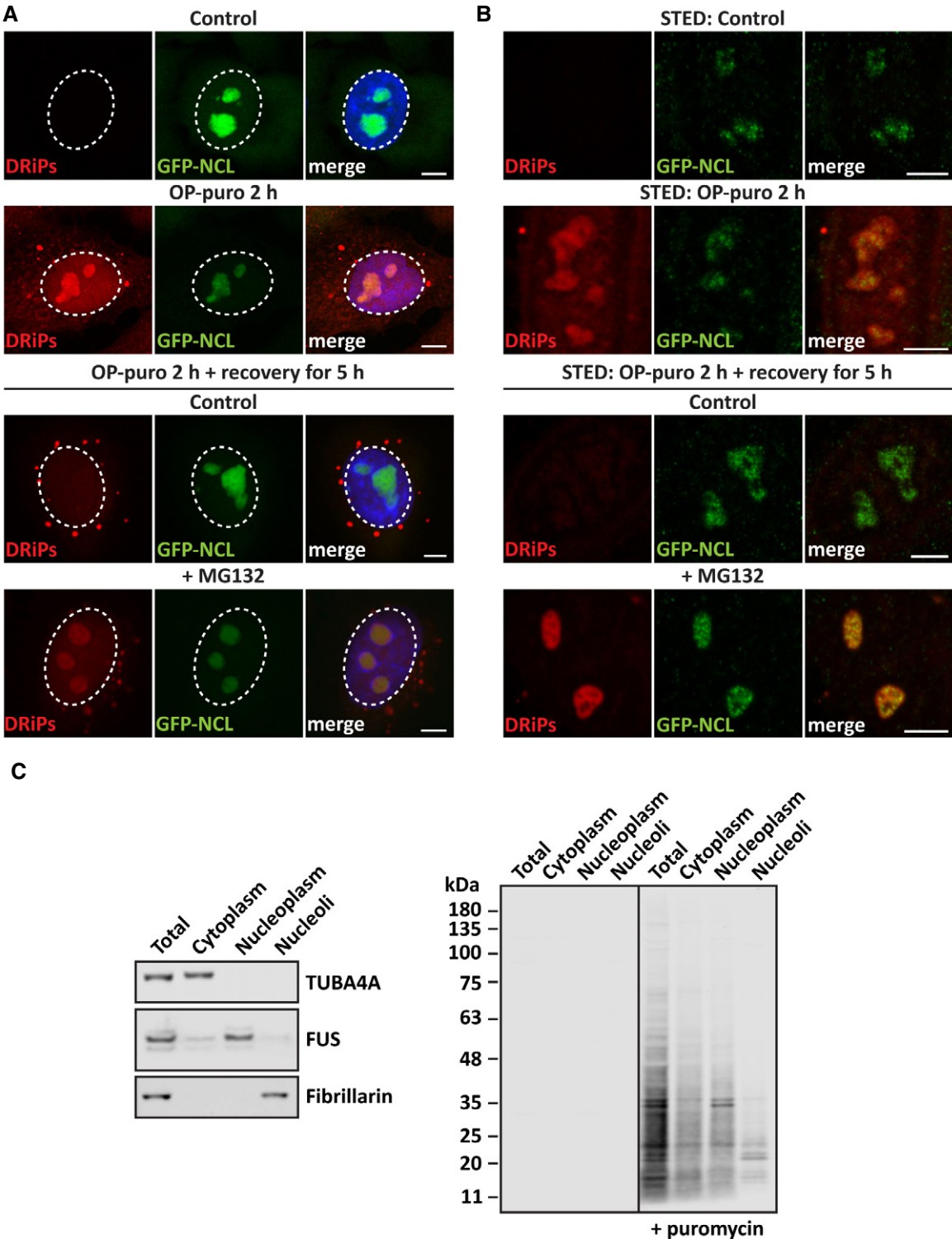

**Figure 1. DRiPs accumulate in nucleoli of normally growing HeLa cells (see also Fig EV1).**

A   DRiP labeling in GFP-NCL HeLa Kyoto cells that were either left untreated or treated with OP-puro (25 μM) for 2 h. Where indicated, cells were allowed to recover in drug-free medium (control) or in presence of MG132 (10 μM) for 5 h. Scale bars: 5 μm.

B   GFP-NCL HeLa Kyoto cells were treated as described in A, with the exception that a lower concentration of OP-puro was used (5 μM). The distribution of DRiPs and GFP-NCL was analyzed by STED super-resolution microscopy. Scale bars: 5 μm.

C   Isolation of nucleoli from HeLa Kyoto cells untreated or treated with puromycin (25 μM) for 2 h. Puromycylated proteins were detected by immunoblotting (right panel). The left panel shows the purity of isolated nucleoli by Western blotting, where fibrillarin was used as nucleolar marker, FUS as nucleoplasmic marker, and TUBA4A as cytoplasmic marker.

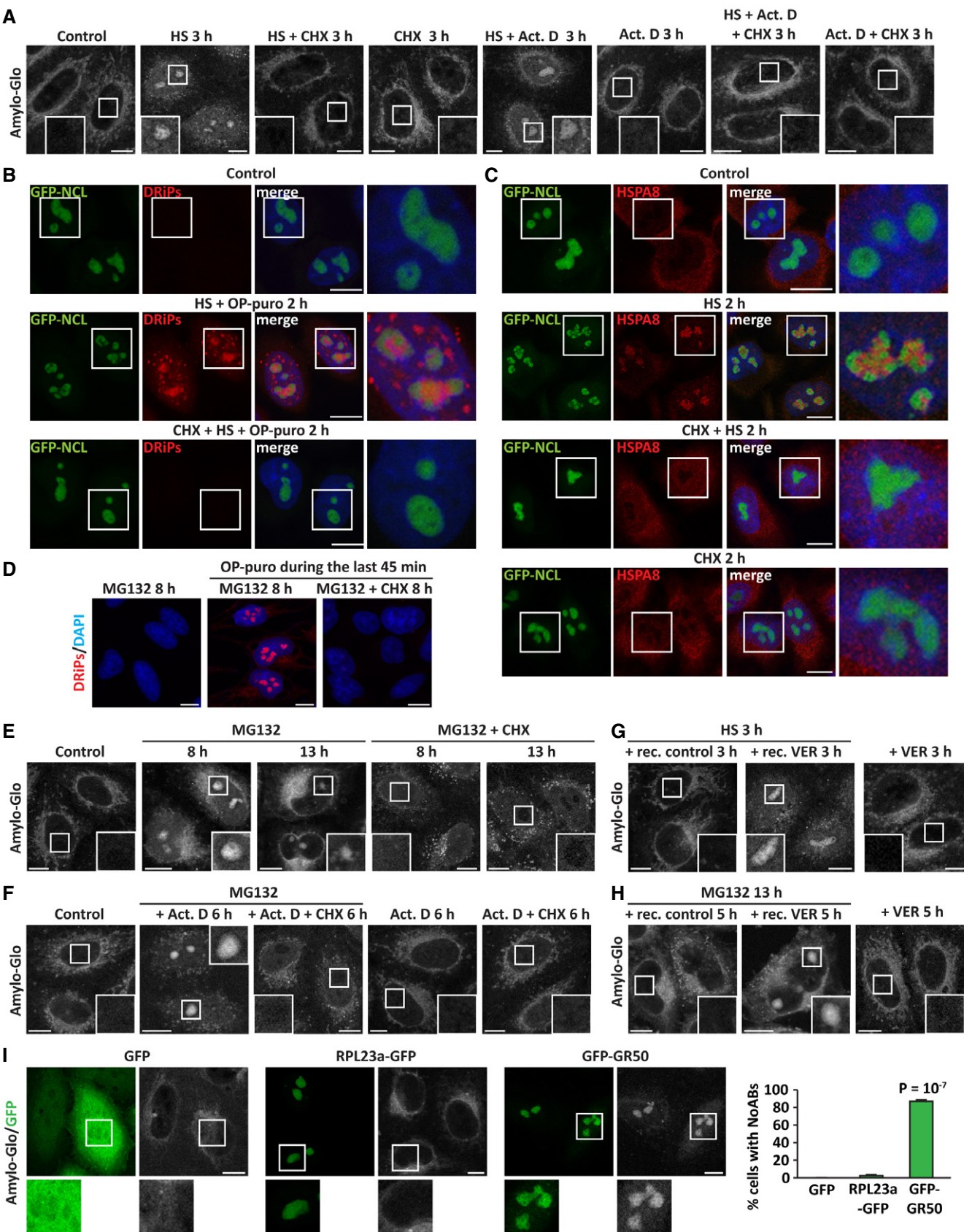

**Figure 2.**

**Figure 2.  Upon heat shock or proteasome inhibition, DRiPs promote amyloidogenesis in NoABs (see also Fig EV2 and Appendix Fig S1).**

A   HeLa cells were left untreated or treated with heat shock (HS) at 42°C for 3 h, alone or with cycloheximide (CHX; 50 μg/ml), or actinomycin D (Act. D; 3 μM) followed by staining with the dye Amylo-glo.

B   DRiP labeling in GFP-NCL HeLa Kyoto cells that were either left untreated or treated with HS at 42°C and OP-puro (25 μM) for 2 h. Where indicated, cells were co-treated with CHX (50 μg/ml).

C   HSPA8 subcellular distribution and recruitment inside nucleoli following exposure to HS at 42°C for 2 h, alone or combined with CHX (50 μg/ml), in GFP-NCL HeLa Kyoto cells.

D   DRiP labeling in HeLa cells treated with MG132 (10 μM) alone or with CHX (50 μg/ml) for 8 h; where indicated, OP-puro (25 μM) was added during the last 45 min, prior to cell fixation.

E   Amylo-glo staining of HeLa cells left untreated or exposed to MG132 (10 μM) alone or combined with CHX (50 μg/ml) for 8 or 13 h.

F   Amylo-glo staining of HeLa cells left untreated or subjected to transcriptional stress (MG132 10 μM and Act. D 4 μM) for 6 h. Where indicated, translation was concomitantly inhibited with CHX (50 μg/ml). Amylo-Glo staining is also shown in HeLa cells treated with Act. D alone or with CHX for 6 h, as control.

G   Amylo-glo staining of HeLa cells exposed to HS at 42°C for 3 h and let to recover for 3 h in drug-free medium (+ rec. control) or with VER (40 μM; + rec. VER). Where indicated, cells were treated with VER alone (VER 3 h).

H   Amylo-glo staining of HeLa cells treated with MG132 (10 μM) for 13 h and let to recover for 5 h in drug-free medium (+ rec. control) or with VER (40 μM; + rec. VER). Where indicated, cells were treated with VER alone (VER 5 h).

I   HeLa cells were transfected with vectors encoding for GFP, GFP-GR50, or RPL23a-GFP. 24 h post-transfection, cells were fixed and stained with Amylo-glo. The percentage of transfected cells with Amylo-Glo-positive nucleoli is shown. Number of cells counted/condition: GFP, 412; RPL23a-GFP, 482; GFP-GR50, 359, in three independent experiments; statistical significance via one-way ANOVA; $P < 10^{-7}$, $\pm$ s.e.m.

Data information: (A–I): Scale bars: 10 μm.

subcompartments (Fig EV2B), nor A-body formation (Fig 2A), while it abrogated the induction of the HSP70/HSPA1A mRNA that occurs upon heat shock (Fig EV2C). These data suggest that endogenous DRiPs rather than pre-existing nuclear proteins that misfold upon heat shock are required for A-body formation, regardless of active transcription.

The A-body was proposed to act as a transient protein storage site that serves an adaptive and protective function during stress. Aside from many other proteins (Scott *et al*, 2010; Jacob *et al*, 2013; Wang *et al*, 2018), A-bodies also recruit HSP70 chaperones, whose ATPase activity is required to orchestrate A-body dissolution once the stress subsides (Audas *et al*, 2016). Indeed, we find that both HSC70/HSPA8 and HSP70/HSPA1A accumulate in the nucleolus upon heat shock (Figs 2C and EV2D). Because HSP70s co-translationally bind to nascent chains (Frydman, 2001; Hartl & Hayer-Hartl, 2002), we next asked whether HSPA8 and HSPA1A are recruited into A-bodies along with DRiPs. During heat shock, inhibition of translation with cycloheximide abrogated DRiP accumulation inside the nucleolus and it also prevented HSPA8 and HSPA1A nucleolar recruitment (Figs 2C and EV2D), while inhibition of transcription with Act. D had no effect (Fig EV2E). Thus, chaperone recruitment into the nucleolus is dependent on active translation and the accumulation of DRiPs.

Second, we studied whether newly synthesized aberrant proteins are also implicated in the formation of the NoA upon prolonged treatment with MG132 (Latonen *et al*, 2011). Similar to heat shock, conversion of nucleoli into an amyloid-like state was observed only when translation was active (Fig 2D and E).

Third, we tested the implication of DRiPs in A-body formation upon transcriptional stress, which is induced by co-treatment of the cells for 6 h with MG132 and Act. D (Audas *et al*, 2016). Transcriptional stress induced the formation of A-bodies (Fig 2F). Translation inhibition during transcriptional stress prevented amyloidogenesis, as well as nucleolar recruitment of HSP70s (Figs 2F and EV2F). Since Act. D treatment alone or combined with proteasome and translation inhibition strongly impaired the synthesis of precursor rRNAs, as measured by qPCR (Appendix Fig S1), our findings demonstrate that DRiPs drive A-body formation independently of nucleolar rRNA synthesis. Based

on the comparable results obtained under different stress conditions, we conclude that the A-body and the NoA are the same structure; therefore, we refer to them as nucleolar amyloid bodies (NoABs).

Taken together, these data demonstrate that upon proteotoxic stress induced by proteasome inhibition or temperature upshift, newly synthesized polypeptide aberrant chains accumulate in nucleolar subcompartments and convert them into an amyloid-like state. Importantly, this amyloidogenesis process is reversible. In fact, when cells were allowed to recover at normal growth temperature or when the proteasome function was restored, NoABs dissolved (Fig 2G and H). Inhibition of the ATPase activity of HSP70 with VER-155008 (VER) during the recovery time prevented the dissolution of NoABs (Fig 2G and H), as previously reported for A-bodies (Audas *et al*, 2016). Thus, NoAB dissolution does not occur spontaneously, but it is assisted by ATP-dependent chaperones that ensure reversibility.

Considering that newly synthesized aberrant polypeptide chains are misfolded (Schubert *et al*, 2000; Verma *et al*, 2013), we tested the impact of aberrant protein accumulation on NoAB formation. To this aim, we used GFP-tagged poly-GR (GFP-GR50). Poly-GR is an unfolded dipeptide repeat that can never adopt a native state and accumulates inside the nucleolus; poly-GR originates from the GGGGCC ($G_4C_2$) repeat expansion in a non-coding region of the C9ORF72 gene, which represents the most common cause of amyotrophic lateral sclerosis (ALS) and frontotemporal dementia (FTD) (Freibaum *et al*, 2015). We found that nucleoli that accumulate GFP-GR50, in the absence of any other type of stress, convert into an amyloid-like state (Fig 2I). By contrast, NoAB did not form in cells overexpressing GFP or the ribosomal protein RPL23a-GFP, which also accumulates inside nucleoli (Fig 2I). Combined these data demonstrate that misfolded proteins play a critical role in nucleolar amyloidogenesis.

## Upon proteotoxic stress, a fraction of DRiPs is transiently sequestered in PML nuclear bodies

Upon heat shock, a pool of DRiPs and newly synthesized proteins also accumulated in nuclear foci distinct from nucleoli (Fig 2B).

These foci did not colocalize with nuclear speckles, where active proteasomal degradation has been reported (Fig EV3A) (Rockel et al, 2005; Scharf et al, 2007). Instead, conventional and STED super-resolution microscopy showed colocalization of DRiPs with PML nuclear bodies (PML-NBs; Fig 3A), which are stress-responsive dynamic membraneless condensates that have been also implicated in proteasomal degradation (Rockel et al, 2005).

Based on these observations, we hypothesized that PML-NBs could facilitate the proteasome-mediated degradation of DRiPs. Most protein degradation by the proteasome requires ubiquitination of polypeptides, with only few exceptions described so far (Verma & Deshaies, 2000; Erales & Coffino, 2014). Indeed, DRiPs accumulating at PML-NBs colocalized with polyubiquitin (polyUb) conjugates in heat-shocked cells (Fig 3A). There are at least two possible explanations for this observation: (i) pre-existing polyUb proteins accumulate at PML-NBs and colocalize with DRiPs; (ii) DRiPs themselves become substrates for ubiquitination. To differentiate between these two possibilities, we inhibited protein synthesis in heat-shocked cells treated with OP-puro. If the polyUb proteins at PML-NBs are pre-existing nuclear proteins that misfold due to temperature upshift, their compartmentalization at PML-NBs would not depend on active translation. Intriguingly, this is not what we found. Accumulation of polyUb at PML-NBs depended on active translation and, conversely, inhibition of translation prevented the sequestration of DRiPs and polyUb conjugates at PML-NBs (Fig 3A), suggesting that DRiPs become ubiquitinated.

These findings support the idea that nuclear accumulation of newly synthesized aberrant proteins activates a specific, yet underappreciated, nuclear stress response. Importantly, this nuclear stress response is not only occurring as a consequence of the forced accumulation of DRiPs in cells treated with OP-puro. In fact, using conventional and STED super-resolution microscopy, we found that heat shock per se, without addition of OP-puro, leads to the transient sequestration of endogenous newly synthesized polyUb conjugates at PML-NBs, which is prevented by concomitant translation inhibition with cycloheximide (Fig 3B–D). We made similar observations for cells exposed to the proteasome inhibitor MG132 (Fig 3E and F), generalizing our findings and suggesting that this nuclear stress response is activated upon different proteotoxic stress conditions. Interestingly, DRiPs sequestered at PML-NBs colocalized with polyUb conjugates, whereas DRiPs that accumulated in NoABs did not (Fig 3E). Similarly, 20S proteasomes only colocalized with DRiPs that accumulated at PML-NBs upon cell treatment with MG132 (Fig 3G), in agreement with previous work reporting proteasomal activity in the nucleoplasm but not in nucleoli (Rockel et al, 2005). These observations point to important differences between nucleoli and PML-NBs and suggest that DRiPs that are compartmentalized at PML-NBs are ubiquitinated for proteasome-mediated degradation. To directly test this hypothesis, we pulled down DRiPs from the nucleoplasmic fraction or from isolated nucleoli using an anti-puromycin antibody and checked DRiP ubiquitination. We compared non-treated cells versus cells treated with puromycin alone or with MG132. Only DRiPs found in the nucleoplasm of cells co-treated with puromycin and the proteasome inhibitor MG132 were ubiquitinated (Fig EV3B). Then, we measured the proteasome-mediated clearance of nuclear DRiPs. After incubation of HeLa cells with puromycin, we allowed the cells to recover in drug-free medium or in the presence of MG132 or ammonium chloride

(NH$_4$Cl), which block proteasome and autophagolysosome-mediated degradation, respectively. Nuclear DRiPs and polyUb conjugates accumulated only when cells recovered in the presence of MG132 (Fig EV3C), further supporting the idea that DRiPs are ubiquitinated and cleared by proteasomes. These results substantiate our microscopy findings and demonstrate that a fraction of nucleoplasmic DRiPs becomes ubiquitinated for proteasome-mediated disposal.

Given that PML is the main component of PML-NBs, we hypothesized that it may act as scaffold for the sequestration of DRiPs. To test this idea, we depleted PML using specific siRNAs (Fig 4A). This significantly decreased the number of PML-NBs (Fig 4B), as well as the number of nuclear foci enriched for DRiPs and polyUb conjugates (Fig 4B and C). Next, we tested the role of SUMOylation in PML body assembly and DRiP compartmentalization by depleting Ubc9, a key enzyme required for SUMOylation of PML (Hay, 2005; Shen et al, 2006). Indeed, Ubc9 depletion impaired PML body assembly (Fig 4D–F) and this correlated with a decrease in the sequestration of DRiPs in nuclear foci (Fig 4G). Hence, we conclude that SUMOylated PML drives PML-NB assembly and DRiP sequestration upon stress.

Finally, we used the model substrate GFP-tagged heat-sensitive luciferase (NLuc-GFP) to generalize the role of PML as a compartment for misfolded proteins (Nollen et al, 2001). We chose NLuc-GFP because it was previously reported to accumulate in nucleoli and in unidentified nuclear foci of cells exposed to heat shock (Nollen et al, 2001). NLuc-GFP was homogeneously distributed in the nucleoplasm of growing unstressed cells; however, following heat shock, NLuc-GFP was sequestered both inside nucleoli and at PML-NBs (Appendix Fig S2). Of note, NLuc-GFP compartmentalization upon stress was largely dependent on active protein synthesis (Appendix Fig S2). During the recovery phase at normal temperature, NLuc-GFP was cleared from nucleoli and PML-NBs (Appendix Fig S2), similar to DRiPs and newly synthesized polyUb proteins. Together, these results establish a role for PML-NBs as stress-responsive overflow compartments that transiently sequester misfolded proteins that accumulate in the nucleus upon proteotoxic stress conditions. In cells that do not express misfolding-prone model proteins such as NLuc-GFP, the majority of these misfolded proteins appear to be DRiPs.

### Chaperones are recruited at PML-NBs to clear newly synthesized misfolded proteins

Previous work demonstrated that PML binds misfolded proteins, such as mutant ataxin-1, and sequesters them into PML-NBs. This, in turn, leads to the recruitment of ubiquitin, E3 ligases, and proteasomes to execute proteolysis of misfolded proteins (Guo et al, 2014). Our data support this model of PML-NBs acting as clearing organelles for misfolded proteins. First, we found that polyUb-DRiPs and proteasomes accumulate at PML-NBs upon proteotoxic stress (Fig 3). Second, we identified additional PQC components recruited at PML-NBs, including the HSP70 chaperones HSPA8 and HSPA1A, and VCP (Appendix Fig S3A–D). HSP70s and VCP bind to DRiPs and newly synthesized proteins and assist their disposal when refolding is not possible (Frydman, 2001; Verma et al, 2013). Thus, we next asked whether these chaperones actively participate in the proteasome-mediated clearance of polyUb-DRiPs from PML-NBs.

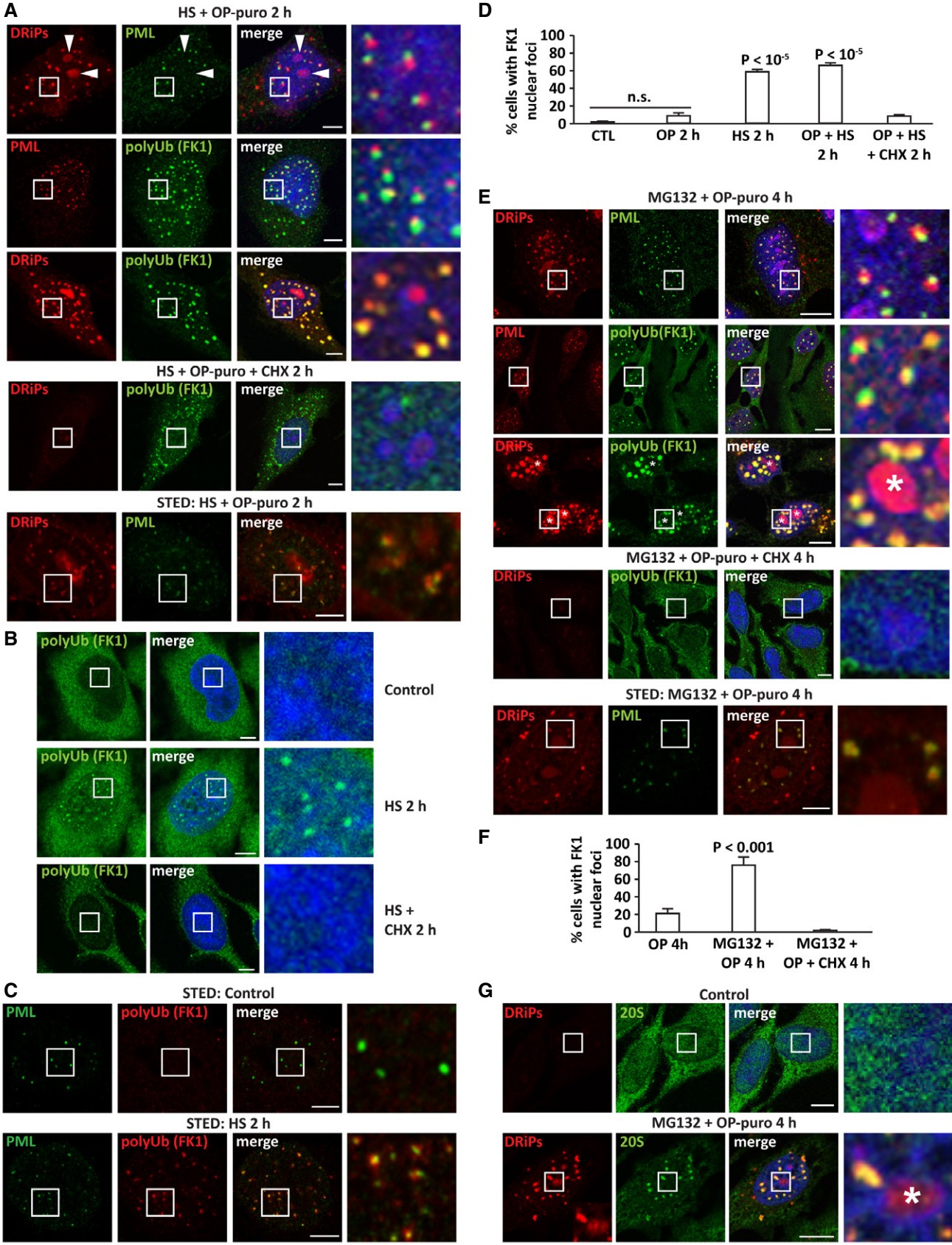

**Figure 3.**

**Figure 3.  Upon heat shock or proteasome inhibition, DRiPs, polyUb proteins and 20S proteasomes accumulate in PML-NBs (see also Fig EV3 and Appendix Fig S2 and Movies EV1–EV3, EV6 and EV7).**

A   DRiPs, polyUb proteins (FK1), and PML labeling in HeLa cells treated with OP-puro (25 μM) and heat shock (HS) at 42°C for 2 h; where indicated, cycloheximide was also added (CHX; 50 μg/ml). Lower panel: STED super-resolution microscopy showing colocalization of DRiPs with PML-NBs in PML-GFP HeLa cells treated with a low concentration of OP-puro (5 μM) and heat shock (HS) at 42°C for 2 h. PML-NBs were visualized using a PML specific antibody, followed by incubation with Alexa Fluor 647. For convenience, the Alexa Fluor 647 signal is shown in green. Scale bars: 5 μm.

B   PolyUb proteins (FK1) labeling in HeLa cells left untreated or treated with HS at 42°C for 2 h, alone or combined with CHX (50 μg/ml). Scale bars: 5 μm.

C   STED super-resolution microscopy showing colocalization of endogenous polyUb proteins (FK1) and PML-NBs in PML-GFP HeLa cells treated with HS at 42°C for 2 h. Scale bars: 5 μm.

D   Quantitation of cells shown in (A–C). Number of cells counted/condition: 712–1,343 in three independent experiments; statistical significance via one-way ANOVA; $P < 10^{-5}$, ± s.e.m.

E   DRiPs, polyUb proteins (FK1), and PML labeling in HeLa cells treated with MG132 (10 μM) and OP-puro (25 μM) for 4 h; where indicated, cycloheximide was also added (CHX; 50 μg/ml). Asterisks (*) indicate nucleoli. Scale bars: 10 μm. Lower panel: STED super-resolution microscopy showing colocalization of DRiPs with PML-NBs in PML-GFP HeLa cells treated with a low concentration of OP-puro (5 μM) and MG132 (10 μM) for 4 h. Scale bars: 5 μm.

F   Quantitation of cells shown in E. Number of cells counted/condition: 298–1,713 in three independent experiments; statistical significance via one-way ANOVA; $P \leq 0.001$, ± s.e.m.

G   Intranuclear distribution of 20S proteasomes in HeLa cells left untreated or treated with MG132 (10 μM) and OP-puro (25 μM) for 4 h. Asterisks (*) indicate nucleoli. Scale bars: 10 μm.

Indeed, inhibition of HSP70s and VCP with VER and Eeyarestatin I (EeyI), respectively, led to the persistence of polyUb conjugates at PML-NBs (Fig 5A–E). These persisting polyUb conjugates colocalized with DRiPs at PML-NBs (Appendix Fig S3E). Similar findings were obtained upon depletion of VCP by siRNA (Fig EV4A–D). We therefore conclude that the clearance of newly synthesized polyUb conjugates from PML-NBs is chaperone-dependent.

### PML-NBs that accumulate misfolded proteins convert into a solid state

PML-NBs are membraneless condensates proposed to form via liquid–liquid phase separation (Banani *et al*, 2016). They are highly dynamic and their number rapidly increases in response to stress (Bernardi & Pandolfi, 2007). Indeed, we found that PML-NB number significantly increased upon various proteotoxic stresses, such as proteasome inhibition or heat shock. Accordingly, the number of PML-NBs decreased to basal levels when cells recovered from stress (Fig EV4E and F). By contrast, restoration of PML-NB number was prevented by inhibition of HSP70s and VCP (Fig EV4E and F). These results suggest that misfolded proteins may directly change the dynamic behavior of PML-NBs and prevent their dissolution, similarly to what we previously observed for SGs in the cytoplasm (Ganassi *et al*, 2016; Mateju *et al*, 2017).

Thus, we next tested whether the accumulation of DRiPs at PML-NBs affects their liquid-like behavior and promotes their transition into a more solid-like state. To this aim, we employed two different approaches, Amylo-Glo staining and FRAP. First, using the Amylo-Glo dye, we found that PML-NBs convert into an amyloid-like state upon proteasome inhibition (Fig 5F). Preventing the accumulation of DRiPs at PML-NBs by co-incubating the cells with cycloheximide prevented amyloidogenesis at PML-NBs. Of note, similar to NoABs (Fig 2), amyloidogenesis at PML-NBs was reversible. In fact, PML-NBs lost their positivity to Amylo-Glo during the recovery from stress, when DRiPs were cleared, but not when DRiP clearance was impaired by inhibition of HSP70 (Fig 5F). Thus, we conclude that the accumulation of DRiPs induces the transient conversion of PML-NBs into an amyloid-like state. Importantly, Amylo-Glo-positive foci overlapping with PML-NBs were also observed in cells exposed to heat shock alone, without addition of OP-puro, further supporting

our interpretation that PML-NBs are stress-responsive overflow compartments that sequester newly synthesized misfolded proteins during stress (Fig 5G).

Second, to test whether misfolded proteins affect PML-NB dynamics, we performed live-cell imaging and FRAP in HeLa cells stably expressing GFP-tagged PML and co-expressing mCherry-VHL, which is a misfolded model protein known to accumulate in the nucleus upon proteotoxic stress (Jacob *et al*, 2013). mCherry-VHL was absent from PML-NBs in resting cells, but it was recruited inside PML-NBs following treatment with MG132 and OP-puro in an active translation-dependent manner, similarly to what we report here for DRiPs (Fig 6A and Movies EV1 and EV2). Using FRAP, we then studied the dynamics of PML-NBs. In control cells, PML-GFP molecules were highly mobile (Fig 6B). By contrast, the mobile fraction of PML-GFP was reduced by > 50% in cells treated with MG132 and OP-puro (Fig 6C). Importantly, PML-NB dynamics were not impaired when cells were co-treated with the translation inhibitor cycloheximide, establishing a causal link between newly synthesized misfolded proteins and PML-GFP mobility within PML-NBs (Fig 6D).

Next, we analyzed PML-GFP mobility during the recovery phase, when mCherry-VHL and DRiPs are cleared from PML-NBs. Misfolded protein disposal from PML-NBs correlated with the restoration of PML-GFP mobility (Fig 6A and E, and Movie EV3). Inhibition of HSP70 during the recovery phase prevented misfolded protein clearance (Fig 6A and Movie EV4) and impaired PML-GFP mobility (Fig 6F). As control, inhibition of HSP70 *per se* did not lead to mCherry-VHL recruitment at PML-NBs, nor did it affect PML-GFP dynamics (Movie EV5). Taken together, our data demonstrate that accumulation of newly synthesized misfolded proteins at PML-NBs reduces their dynamic behavior and promotes their maturation into a solid amyloid-like state. Chaperones, such as HSP70s, revert this process and restore PML-NB dynamics upon recovery from stress.

### Solid PML-NBs sequester nuclear ubiquitin, chaperones, and 20S proteasomes

Our data show that DRiP-containing PML-NBs sequester a large fraction of nuclear ubiquitin, as well as 20S proteasomes, HSP70s, and VCP (Fig 3E and G, Appendix Fig S3A–D). We thus asked

whether these PQC factors become immobilized when PML-NBs convert into the solid amyloid-like state. Ubiquitin, 20S proteasomes, HSP70s, and VCP were all released from PML-NBs when DRiPs were efficiently cleared in recovering cells (Fig 6G). By contrast, when the clearance of DRiPs from PML-NBs was impaired

due to HSP70, VCP, or proteasome inhibition, these PQC factors remained sequestered together with undigested DRiPs at PML-NBs (Fig 6G and Appendix Fig S3E). We next used HeLa cells stably expressing the GFP-tagged proteasome subunit alpha 7 (GFP-PSMA7) to determine the mobility of proteasomes (Poser *et al*,

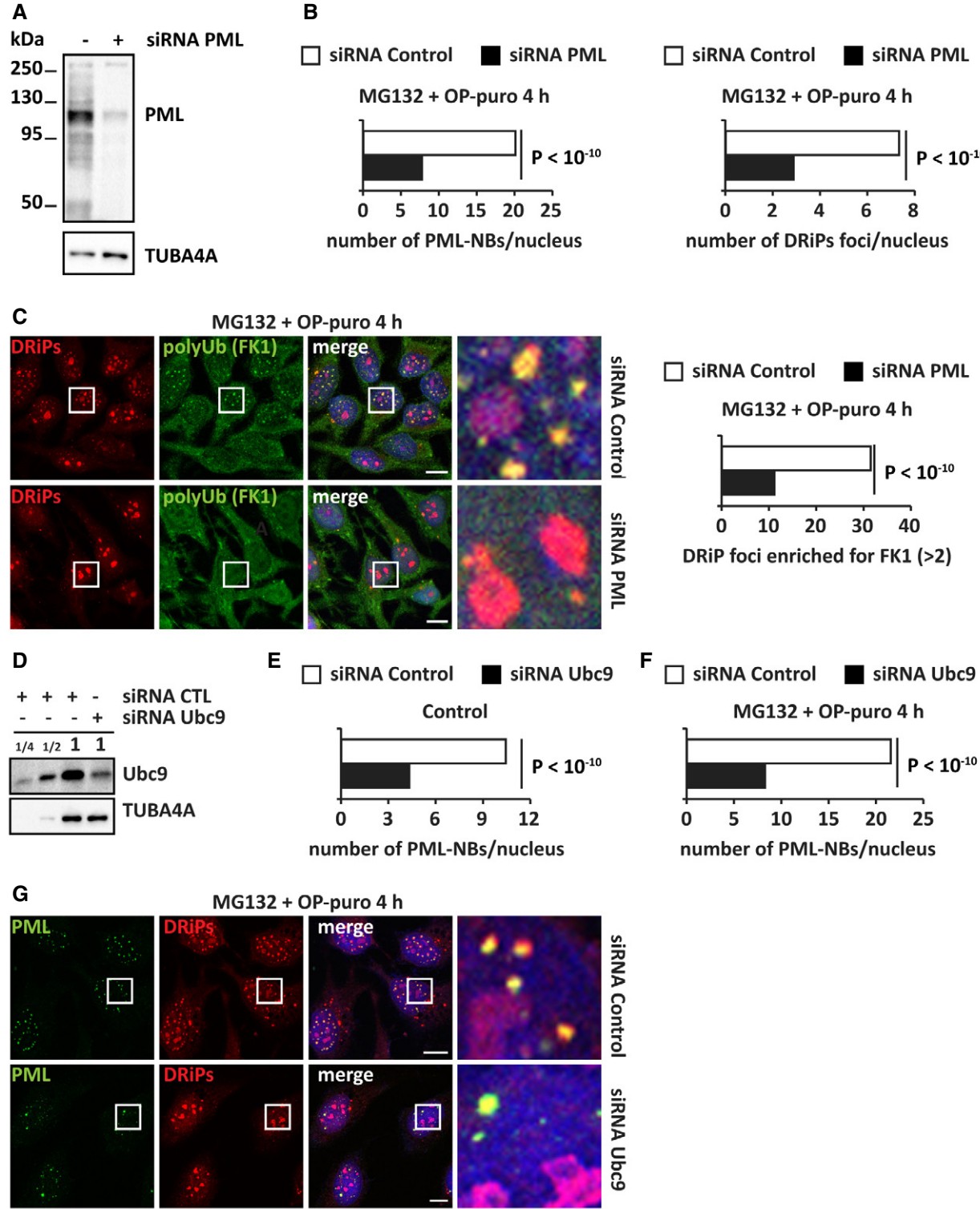

**Figure 4.**

◀

**Figure 4. Compartmentalization of DRiPs and polyUb proteins is catalyzed by the self-assembly of PML into nuclear bodies.**

A HeLa cells were lipofected with non-targeting siRNA control or a specific siRNA against PML. 72 h post-transfection, total proteins were extracted and expression levels of PML were analyzed by immunoblotting. TUBA4A was used as loading control.

B HeLa cells were lipofected with non-targeting siRNA control or a specific siRNA against PML. 72 h post-transfection, cells were treated with MG132 (10 µM) and OP-puro (25 µM) for 4 h. Left panel: Automated quantitation of the number of PML-NBs/nucleus is reported. Statistical significance via 2-tailed Student's $t$-test. Cell number analyzed: 334–364; $P < 10^{-10}$. Right panel: automated quantitation of the number of DRiP foci/nucleus. Cell number analyzed: 334–364; $P < 10^{-10}$. Automated DRiP segmentation is based on azide signal.

C DRiPs and polyUb (FK1) protein distribution in cells treated as described in (A). Automated DRiP segmentation is based on azide signal. Quantitation of the number of DRiP foci enriched for polyUb (> 2) is shown; $n$ = 590–2,114; statistical significance via 2-tailed Student's $t$-test; $P < 10^{-10}$. Scale bars: 10 µm.

D Immunoblotting on protein extracts from HeLa cells lipofected with non-targeting siRNA control or a specific siRNA against Ubc9. A dilution curve of the control sample is shown. TUBA4A was used as loading control.

E Quantitation of the number of PML-NBs/nucleus from cells treated as described in (D). Statistical significance via 2-tailed Student's $t$-test. Cell number analyzed: 192–430; $P < 10^{-10}$. Automated PML-NB segmentation is based on PML signal.

F Cells lipofected for 72 h with non-targeting siRNA control or a specific siRNA against Ubc9 were treated with MG132 (10 µM) and OP-puro (25 µM) for 4 h. Automated quantitation of the number of PML-NBs/nucleus is reported. Statistical significance via 2-tailed Student's $t$-test. Cell number analyzed: 151–225; $P < 10^{-10}$ (see panel G).

G Representative pictures showing that Ubc9-depleted cells cannot efficiently induce the self-assembly of PML-NBs upon treatment with MG132 (10 µM) and OP-puro (25 µM) for 4 h. Scale bars: 10 µm.

2008). Using FRAP, we measured the fraction of fast, slow, and immobile GFP-PSMA7 molecules in resting and stressed cells. In resting cells, GFP-PSMA7 was diffusely distributed in the nucleoplasm and was highly mobile (Movie EV6, Fig 6H and Fig EV5A). Upon treatment of the cells with MG132 and OP-puro, GFP-PSMA7 was recruited, together with the misfolding model protein mCherry-VHL, at PML-NBs; this correlated with a significant decrease in GFP-PSMA7 mobility (Movie EV7, Fig 6H and Fig EV5A). During the stress recovery phase, GFP-PSMA7 molecules were released and their mobility was restored (Movie EV8, Fig 6H and Fig EV5A). However, upon inhibition of HSP70s with VER, GFP-PSMA7 lost its dynamic behavior and became sequestered in solid amyloid-like PML-NBs (Movie EV9, Fig 6H and Fig EV5A). These data suggest that HSP70 prevents the irreversible aggregation of proteasomes at PML-NBs during acute stress and helps to restore the dynamics of proteasomes upon stress relief.

## Solid PML-NBs deplete ubiquitin and delay the relocalization of 53BP1 at fragile chromosomal sites

Nuclear proteostasis maintenance is fundamental for the preservation of genome integrity (Shibata & Morimoto, 2014). Physiological proliferation of mammalian cells, which have gigabase-sized genomes, inevitably implies some degree of DNA replication failure and spontaneous DNA or chromatin lesions. At each cell cycle, portions of the DNA remain unreplicated, leading to the formation of double fork stalls that are marked by the DNA repair protein p53-binding protein 1 (53BP1) and appear as 53BP1-positive foci (Harrigan *et al*, 2011; Lukas *et al*, 2011). DNA replication and DNA damage sensing/repair are dependent on ubiquitin, proteasomes, VCP, and PML, the very same factors that we identified as essential for DRiP compartmentalization and clearance. For example, the DNA replication machinery is regulated by VCP in a ubiquitin-dependent manner (Deichsel *et al*, 2009; Meerang *et al*, 2011) and histone H2A ubiquitination is required for proper functioning of DNA repair factors, including 53BP1 (Krogan *et al*, 2004; Doil *et al*, 2009; Mattiroli *et al*, 2012; Thorslund *et al*, 2015; Schwertman *et al*, 2016). Based on these findings, we asked whether, by reducing the amounts of soluble ubiquitin, 20S proteasomes, or VCP, solid PML-NBs may affect DNA damage sensing/repair and the formation of 53BP1 foci.

In agreement with the literature, proliferating HeLa cells contained nuclear 53BP1 foci that colocalized with ubiquitinated H2A (H2A-Ub; Fig EV5B; Harrigan *et al*, 2011). As previously shown by independent groups, we also found that the number of these 53BP1 foci strongly decreased upon proteasome inhibition with MG132, alone or combined with OP-puro (Fig 7A and B), and this, in turn, correlated with a reduction of H2A-Ub levels and an accumulation of polyUb substrates (Fig EV5C; Mimnaugh *et al*, 1997; Jacquemont & Taniguchi, 2007; Mailand *et al*, 2007; Chroma *et al*, 2016). The number of 53BP1 foci returned to basal levels when proteasomal activity was restored during the recovery phase, indicating that nuclei recovered their ability to recognize and protect fragile chromosomal sites from erosion (Fig 7A and B). Importantly, upon MG132 treatment, polyUb-DRiPs were sequestered with 20S at PML-NBs; instead, upon treatment of the cells with OP-puro alone we did not observe accumulation of ubiquitin inside nucleoli, and the pool of DRiPs that was transiently compartmentalized in nucleolar subcompartments was rapidly cleared by proteasomes. In agreement, treatment of the cells with OP-puro alone did not significantly affect 53BP1 foci formation (Fig 7B). Together, these observations suggest that different processes in the nucleus compete for a limiting pool of ubiquitin. In cells exposed to proteotoxic stress, nuclear ubiquitin mainly serves to label misfolded proteins for their subsequent clearance at the expense of DNA damage sensing/repair. When normal proteostasis in the nucleus is restored, ubiquitin is primarily used to orchestrate the process of DNA damage sensing/repair. This suggests that there is a trade-off between nuclear proteostasis and genome integrity and that stressed cells have to balance these two processes in order to remain healthy.

Since newly synthetized proteins are the main substrate for polyubiquitination upon stress (Fig 3B and E), we reasoned that inhibition of translation during proteotoxic stress should increase the ubiquitin pool available for DNA damage sensing/repair. In fact, translation inhibition during stress rescued the formation of 53BP1 foci that colocalized with H2A-Ub (Fig EV5B and D). Accordingly, H2A-Ub levels were also partly restored (Fig EV5C). As control, cycloheximide *per se* had no effect on 53BP1 foci formation, nor H2A-Ub levels (Fig EV5B–D). Next, we determined whether delaying the clearance of DRiPs during the stress recovery phase impairs DNA damage sensing. Indeed, we found that inhibition of HSP70s

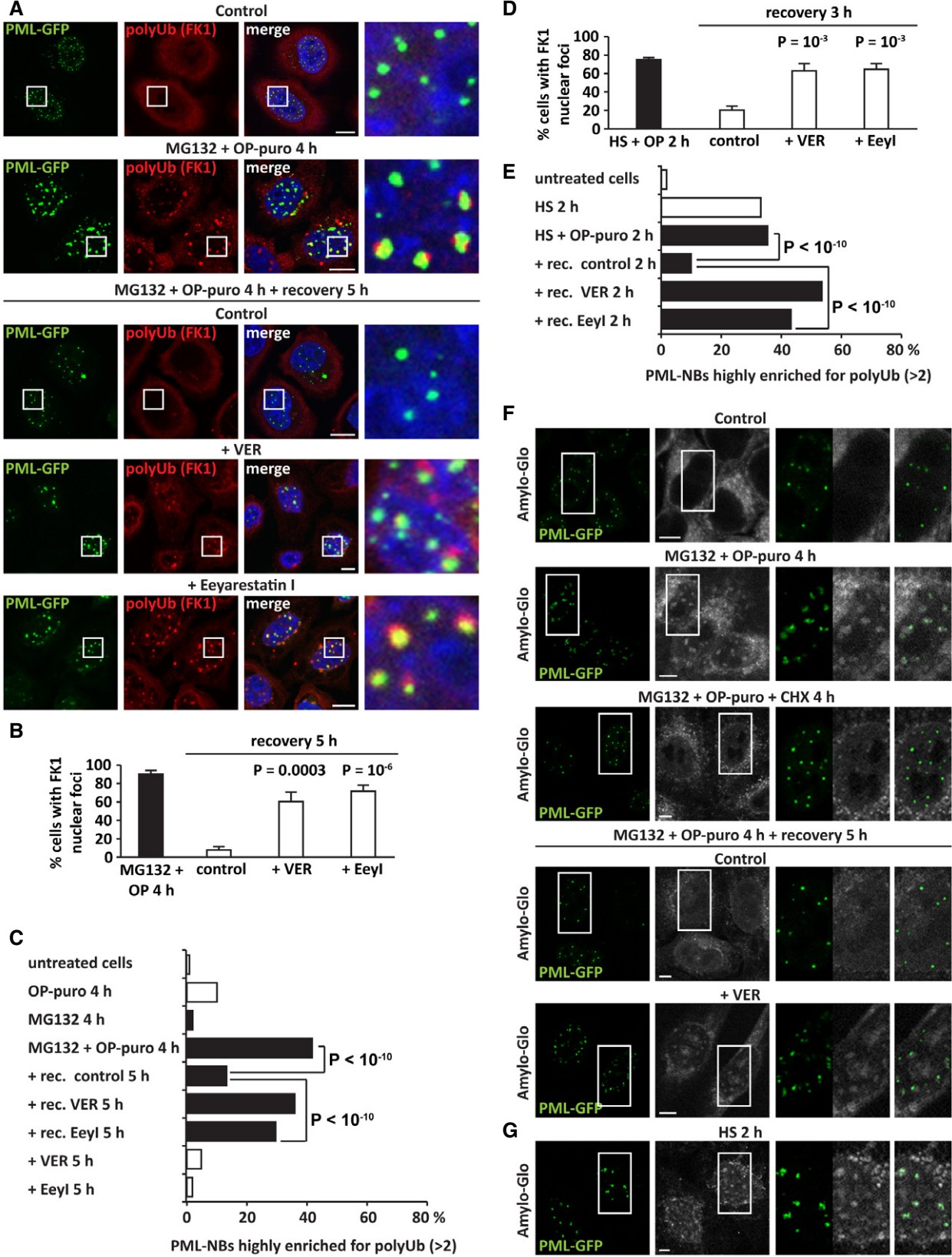

**Figure 5.**

◄

**Figure 5. PolyUb proteins are cleared from PML-NBs by chaperones (see also Appendix Figs S3 and EV4 and Movies EV1–EV5).**

A   PolyUb protein (FK1) labeling in PML-GFP HeLa Kyoto cells that were either left untreated or treated with MG132 (10 μM) and OP-puro (25 μM) for 4 h. Cells were either immediately fixed or let to recover for 5 h in drug-free medium (Control), with VER (40 μM) or Eeyarestatin I (5 μM). Scale bars: 10 μm.

B   Quantitation of the % of cells with polyUb (FK1) nuclear foci shown in (A). Number of cells counted/condition: 310–1,258 in three independent experiments; statistical significance via one-way ANOVA; $P = 0.0003$ or $10^{-6}$, $\pm$ s.e.m.

C   Quantitation of the number of PML-NBs enriched for polyUb (> 2) in cells treated as described; $n = 611$–2,612; statistical significance via one-way ANOVA; $P < 10^{-10}$. Automated PML-NB segmentation is based on PML signal.

D   Quantitation of the % of cells with polyUb proteins (FK1) sequestered in nuclear foci. HeLa cells were treated with OP-puro (25 μM) and heat shock (HS) at 42°C for 2 h. Then, cells were either immediately fixed or let to recover for 3 h in drug-free medium (control), with VER (40 μM) or Eeyarestatin I (5 μM). Number of cells counted/condition: 593–1,016 in three independent experiments; statistical significance via one-way ANOVA; $P = 10^{-3}$, $\pm$ s.e.m.

E   Quantitation of the number of PML-NBs enriched for polyUb (> 2) in HeLa cells treated as described in (D) but with a recovery time of 2 h; $n = 1,155$–3,283; statistical significance via one-way ANOVA; $P < 10^{-10}$. Automated PML-NB segmentation is based on PML signal.

F   Amylo-Glo staining of PML-GFP HeLa Kyoto cells that were subjected to the following treatments: untreated (Control), MG132 (10 μM), and OP-puro (25 μM) for 4 h, alone or combined with CHX (50 μg/ml). Where indicated, cells were allowed to recover for 5 h in drug-free medium (Control) or with VER (40 μM). Confocal images showing Amylo-Glo and PML-GFP. Scale bars: 5 μm.

G   Amylo-Glo staining of PML-GFP HeLa Kyoto cells that were subjected to heat shock (HS) at 42°C for 2 h. Confocal images showing Amylo-Glo and PML-GFP. Scale bars: 5 μm.

and VCP after proteotoxic stress impaired 53BP1 foci formation, while inhibition of HSP70 and VCP alone had no effect (Fig 7A and B). We conclude that inhibiting the clearance of newly synthetized and polyUb-DRiPs limits the availability of ubiquitin for H2A ubiquitination, thus affecting DNA damage sensing.

To further validate our interpretation, we tested whether the restoration of a normal soluble pool of ubiquitin or VCP could rescue 53BP1 recruitment to fragile chromosomal sites. Overexpression of Flag-VCP could not rescue 53BP1 foci formation upon proteasome inhibition and OP-puro treatment (Fig 7C and D), indicating that it is not a limiting factor. By contrast, overexpression of ubiquitin fully restored 53BP1 foci formation in stressed cells, while no effect was seen in untreated cells (Fig 7C and D). Altogether, these data demonstrate that solid PML-NBs accumulate newly synthetized polyUb-DRiPs and immobilize nuclear ubiquitin, along with chaperones and 20S proteasomes. This depletion of nuclear ubiquitin, in turn, compromises the formation of DNA repair compartments at fragile chromosomal sites.

**Prolonged immobilization of ubiquitin at solid PML-NBs decreases cell fitness**

If not properly resolved, DNA replication stress causes DNA damage and promotes genome instability; this, in turn, decreases cell proliferation (Zhang *et al*, 2013) and ultimately contributes to aging and human diseases such as cancer and neurodegeneration (Zeman & Cimprich, 2014). The spontaneous fragile chromosomal sites that are shielded from erosion by 53BP1 foci are normally transmitted to daughter cells for subsequent repair, enabling proper cell proliferation (Lukas *et al*, 2011). Thus, we next asked whether the immobilization of ubiquitin, 20S, and chaperones at solid PML-NBs affects cell proliferation.

We assessed cell proliferation by colony formation assays, counting the surviving colonies 10 days after exposure to proteotoxic stress. First, we studied HeLa cells exposed to MG132 and OP-puro (or puromycin) for 4 h, followed by recovery in drug-free medium. These cells showed 53BP1 foci (Fig EV5E) and displayed no significant impairment in cell proliferation (Fig 7E). Second, we counted the number of colonies surviving upon HSP70s and VCP inhibition. Cells treated with VER or EeyI alone that recovered in drug-free medium also formed 53BP1 foci (Fig EV5E) and showed no

significant proliferation impairment (Fig 7E). By contrast, cell proliferation was compromised when HSP70s and VCP were inhibited during the recovery from proteotoxic stress (Figs 7E and EV5E). We noticed that the impact of VCP inhibition on cell proliferation was stronger than the one of HSP70 inhibition. HSP70 inhibition delayed the clearance of misfolded proteins (Fig 5) and the sensing and shielding of fragile chromosomal sites (Fig 7B), strongly decreasing cell proliferation (Figs 7E and EV5E). Instead, VCP inhibition provoked the sudden death of the majority of the cells. The few remaining cells were characterized by undigested DRiPs and diffuse 53BP1 staining (Fig EV5E), and they completely lost their ability to proliferate and form colonies after 10 days (Fig 7E). We conclude that the proper clearance of polyUb-DRiPs from PML-NBs is critical for maintaining cell proliferation. Thus, the transient sequestration of DRiPs and PQC factors at PML-NBs, although temporarily affecting their dynamics, represents a protective stress response. By contrast, failure to restore normal proteostasis and free ubiquitin levels, as it may occur during severe stress, can compromise genome health and cell fitness.

# Discussion

In this work, we show that a large fraction of newly synthesized defective ribosomal products (DRiPs) diffuses into the nucleus. Here, DRiPs are compartmentalized in nucleolar bodies and PML-NBs for later disposal by chaperones and proteasomes. If not efficiently cleared, DRiPs change the dynamic state of nucleolar bodies and PML-NBs. Specifically, DRiPs lead to amyloidogenesis of nucleolar bodies, generating the NoABs, and induce the transition of PML-NBs into a solid amyloid-like state. The delayed dissolution of solid PML-NBs, by sequestering 20S proteasomes and limiting the soluble pool of free ubiquitin, compromises the formation of DNA repair compartments at fragile chromosomal sites, with severe consequences for genome health and cell viability. Hence, nuclear proteostasis and genome integrity are two competing processes that have to be well balanced to keep a cell in a healthy state.

What are the DRiPs that accumulate in the nucleus of stressed cells? And how abundant are they in normally growing cells? Our finding that exposure of HeLa cells to heat shock *per se*, without addition of OP-puro, activates the formation of NoABs and the

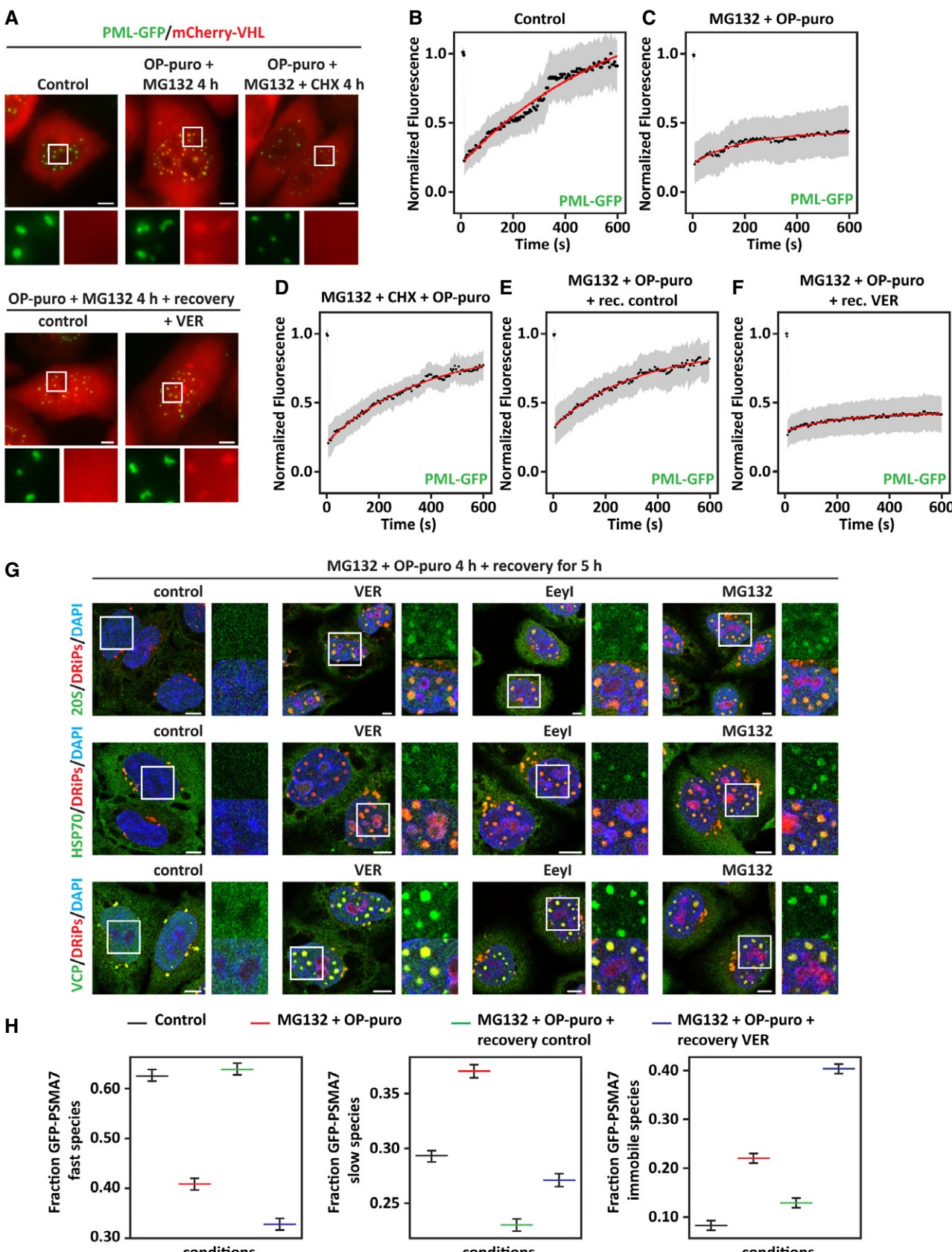

**Figure 6.**

◄

**Figure 6.   DRiPs and mCherry-VHL convert PML-NBs into an amyloid solid-like state that chaperones can revert (see also Fig EV4 and Movies EV1–EV9).**

A      PML-GFP HeLa Kyoto cells were lipofected with a cDNA encoding for mCherry-VHL. 24 h post-transfection, cells were left untreated or treated with MG132 (10 μM) and OP-puro (25 μM) for 4 h. Where indicated (OP-puro + MG132 + CHX), translation was inhibited during stress with cycloheximide (CHX; 50 μg/ml). In the lower panel, cells were allowed to recover for 5 h in drug-free medium (Control) or with VER (40 μM). The subcellular distribution of PML-GFP and mCherry-VHL was studied by live-cell confocal imaging. Representative pictures are shown. Scale bars: 5 μm.

B–F    Quantitation of the fluorescence intensity recovery after bleach of PML-GFP. PML-GFP HeLa Kyoto overexpressing mCherry-VHL for 24 h and treated as described in (A) were analyzed. The mean of 19–23 FRAP curves and the fitting curves are shown in black and red, respectively. In gray, the SD is shown.

G      Intranuclear distribution of 20S proteasomes, HSP70, VCP, and DRiPs in HeLa cells treated with MG132 (10 μM) and OP-puro (25 μM) for 4 h and allowed to recover for 5 h in drug-free medium (control), with VER (40 μM), Eeyarestatin I (5 μM), or MG132 (10 μM). Scale bars: 5 μm.

H      GFP-PSMA7 HeLa Kyoto cells were left untreated or treated as described in A. Cells were subjected to FRAP to analyze GFP-PSMA7 mobility. Quantitation of three populations of GFP-PSMA7 molecules is shown: fast moving, slow moving, and immobile; statistical significance via one-way ANOVA; $n = 12$–$14 \pm$ SEM.

compartmentalization of newly synthesized polyUb proteins at PML-NBs (Figs 2 and 3) supports the interpretation that these species are abundant and represent a potential threat to the nuclear proteome. However, detection of these endogenous DRiPs is technically difficult and would require the development of sophisticated proteomic tools. There are multiple possible sources for endogenous DRiPs. First, DRiPs could be generated by pervasive translation outside of annotated coding regions, especially in 5′ UTRs and lncRNAs. Ribosome profiling recently identified pervasive ribosome occupancy in these regions and provided evidence for their translation into low molecular weight products, which most likely are not properly folded (Ingolia et al, 2014). Second, it is also conceivable that a fraction of DRiPs that accumulate in NoABs and PML-NBs originate from biosynthetic errors, as well as damaged mRNAs. Third, it was recently proposed that many ribosomes prematurely abort translation, generating short peptides containing few amino acids (Baboo et al, 2014). All these aberrant products should be sufficiently small to rapidly diffuse into the nucleus. Finally, some DRiPs may also be generated via non-canonical RAN translation of repeated sequences, whose expansion is associated with an increasing number of neurodegenerative diseases, including polyglutamine diseases, myotonic dystrophy type 1, and amyotrophic lateral sclerosis (ALS) (Cleary & Ranum, 2017). In line, we find that ALS-linked GFP-GR50, which was previously reported to accumulate in nucleolar subcompartments (Lee et al, 2016), promotes nucleolar amyloidogenesis in the absence of additional stress. This suggests that NoAB formation could contribute to disease-associated toxicity together with changes in nucleolar functions and nucleocytoplasmic transport, which were previously documented (Kwon et al, 2014; Freibaum et al, 2015). Collectively, all these aberrant peptides may compete for chaperone-assisted folding and quality control with newly synthesized proteins that have not yet reached their native state, as well as with supersaturated and metastable proteins, which are intrinsically prone to aggregation (Ciryam et al, 2013, 2015). This, in turn, could poison the proteome and promote cytoplasmic and nuclear protein aggregation, especially in cells that experience additional stress. Thus, conditions that impair the functionality of the cytoplasmic protein quality control and ribosomal quality control systems could increase the load of DRiPs, favoring their accumulation inside the nucleus and, ultimately, compromising genome integrity and cell viability.

In normally growing HeLa cells, DRiPs are compartmentalized in nucleolar bodies, without affecting nucleolar dynamics and function. With time, DRiPs are cleared from nucleolar bodies by proteasomes. We, therefore, refer to nucleolar bodies as constitutive overflow compartments for DRiPs. However, when the total amount

of these substrates exceeds the nuclear proteostasis capacity, due to proteasome inhibition or temperature upshift, nucleolar bodies turn into an amyloidogenic state that can only be reverted by the action of chaperones. Similar structures were previously reported in cells exposed to heat shock, acidosis, transcriptional stress, or prolonged proteasome inhibition and were called A-bodies or nucleolar aggresomes, respectively (Latonen et al, 2011; Audas et al, 2016). Here, we demonstrate that these are the same structure with amyloidogenic properties, which we call NoABs. Although we cannot exclude that native proteins, such as ribosomal proteins and proteins with nucleolar localization signals, may contribute to protein aggregation (Scott et al, 2010; Audas et al, 2012, 2016; Jacob et al, 2013), our data demonstrate the fundamental importance of newly synthesized misfolded proteins for amyloidogenesis of NoABs, regardless of active transcription.

PML-NBs exist in cells grown under physiological conditions. However, upon stress, PML-NBs grow substantially in size and number. This adaptation requires the assembly of PML-NBs via liquid–liquid demixing of SUMOylated PML (Banani et al, 2016) and is dependent on the accumulation of misfolded DRiPs. When the stress subsides, HSP70s and VCP promote the release of misfolded proteins from PML-NBs and their proteasome-mediated degradation. Subsequently, PML-NBs disassemble and their number decreases back to physiological levels. These findings suggest that PML fulfills distinct functions in resting and stressed cells. In resting cells, PML mainly exerts housekeeping functions in post-translational control, such as SUMOylation, epigenetic control, and chromatin dynamics (Lallemand-Breitenbach & de The, 2018). However, when the concentration of newly synthesized misfolded proteins reaches a critical threshold in the nucleoplasm, for example during stress, PML-NBs act as stress-responsive overflow compartments that recruit these potentially toxic misfolded proteins. This shows that PML-NBs are adaptable compartments that recruit a range of different substrates depending on the cellular conditions.

The accumulation of misfolded proteins at PML-NBs can promote their conversion from a liquid-like state, where PML molecules are highly mobile, into a solid amyloid-like state, where PML molecules are almost immobile. Importantly, this conversion can be reversed by chaperones, such as HSP70s and VCP, once the stress subsides. When this process fails, PML-NBs keep the solid amyloid-like properties and immobilize chaperones, ubiquitin, and 20S proteasomes, compromising their nuclear functions. We previously described a similar mechanism affecting the liquid-like properties of SGs. Thus, nucleoli and PML-NBs represent other examples of membraneless organelles that are vulnerable to protein misfolding and,

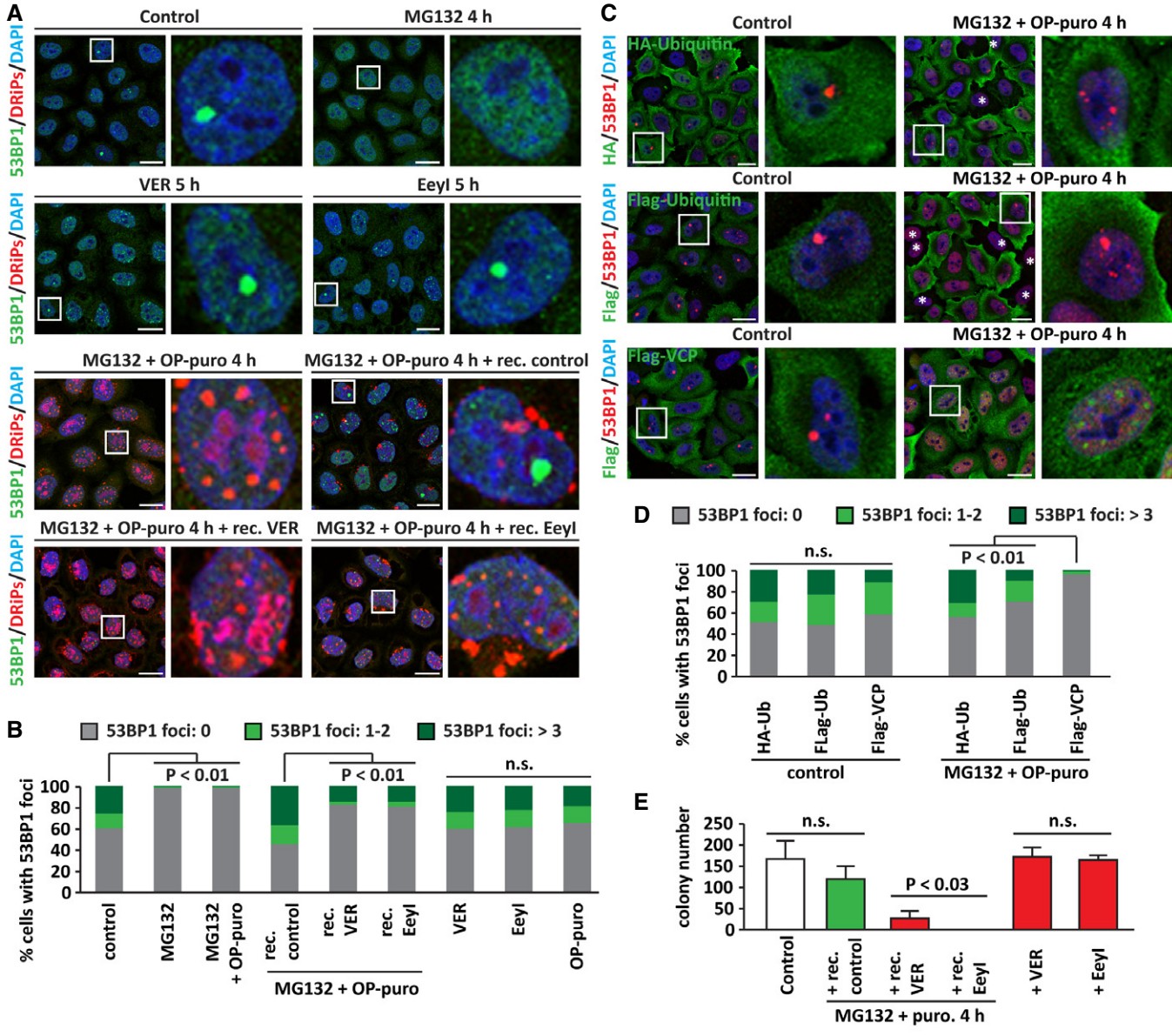

**Figure 7. Sequestration of ubiquitin in solid PML-NBs impairs the recruitment of 53BP1 at fragile chromosomal sites (see also Fig EV5).**

A   Subcellular distribution of 53BP1 and DRiPs in HeLa cells that were left untreated or treated with MG132 (10 μM) and OP-puro (25 μM) for 4 h, followed by 5 h recovery in drug-free medium (+ rec. control), with VER (40 μM; + rec. VER) or Eeyarestatin I (5 μM; + rec. EeyI). Where indicated, cells were treated with MG132, VER or EeyI alone for 5 h. Scale bars: 20 μm.

B   Quantitation of the % of cells with 53BP1 foci. Cells were treated as described in (A) and divided into three categories, based on the number of 53BP1 foci/nucleus: 0, 1–2, and > 3. Number of cells counted/condition: 1,112–1,537 in three independent experiments; statistical significance via one-way ANOVA; *P* < 0.01.

C   HeLa cells were lipofected with cDNAs encoding for HA-Ubiquitin, Flag-Ubiquitin, or Flag-VCP. 24 h post-transfection, cells were either left untreated or treated with MG132 (10 μM) and OP-puro (25 μM). Cells were processed for staining of HA, Flag, and endogenous 53BP1. * indicates cells that do not express HA- or Flag-Ubiquitin and show no rescue of 53BP1 foci formation after stress. Scale bars: 20 μm.

D   Quantitation of the % of cells with 53BP1 foci. Cells were treated as described in (C) and divided into three categories, based on the number of 53BP1 foci/nucleus: 0, 1–2, and > 3. Number of cells counted/condition: 710–938 in three independent experiments; statistical significance via one-way ANOVA; *P* < 0.01.

E   Colony formation assay. The number of colonies surviving 10 days after plating is shown. Control cells or cells treated with MG132 (10 μM) and puromycin (5 μg/ml), followed by 5 h recovery in drug-free medium (+ rec. control), with VER (40 μM; + rec. VER) or Eeyarestatin I (5 μM; + rec. EeyI) are shown. Where indicated, cells were treated with VER or EeyI alone for 5 h. Number of colonies counted/condition: up to 526 in three independent experiments; statistical significance via one-way ANOVA; *P* < 0.03, ± s.e.m.

particularly, to DRiPs (Ganassi *et al*, 2016; Mateju *et al*, 2017). Altogether, changes in the dynamics of membraneless organelles seem to be a major driver of cell dysfunction and disease.

What is the functional role of DRiP sequestration in NoABs and PML-NBs? Our data suggest that the compartmentalization of DRiPs could represent a protective mechanism serving at least two

functions: first, to avoid the promiscuous interaction of DRiPs with essential nuclear components; second, to promote the efficient clearance of DRiPs by concentrating ubiquitin, 20S proteasomes, and chaperones at specific PQC sites. Although we have not studied where proteasomes clear DRiPs, it is likely that DRiP degradation occurs close to PML-NBs, where the concentrations of polyubiquitinated substrates and 20S proteasomes are high. We also detected proteasomes inside NoABs upon severe stress conditions, such as prolonged inhibition of chaperones (Fig EV5A), suggesting that some degradation of DRiPs may also occur inside NoABs. However, the absence of ubiquitinated DRiPs and proteasomes in nucleoli of cells exposed to milder stress conditions suggests that this is not the major degradation site of nuclear DRiPs.

Stress conditions constitute a major threat to cellular health, in part because they increase the amount of DRiPs. Cells respond by arresting the cell cycle (Kuhl & Rensing, 2000; Shibata & Morimoto, 2014) and by turning on the sequestration of DRiPs in nucleoli and PML-NBs, where the PQC machinery is recruited for later DRiP disposal. Cell cycle arrest and DRiP sequestration promote cellular adaptation to stress, but these processes have to be well coordinated in order to ensure cell survival. This is because two essential cellular processes—nuclear proteostasis and DNA damage repair—compete for the same components. Thus, if not properly handled, ubiquitinated newly synthesized proteins and ubiquitinated nucleoplasmic DRiPs limit the available pool of ubiquitin, potentially threatening genome stability.

In summary, our data highlight an intricate connection between membraneless compartments and protein misfolding, lending further support to the idea that aberrant changes in the dynamics of membraneless organelles can drive cell dysfunction and disease (Alberti & Carra, 2018). Our data also identify nuclear protein quality control as a weak link in the proteostasis network, because of the continuous targeting of newly synthesized aberrant proteins to the nucleus for proteasomal degradation. Upon proteasome inhibition, a link between decreased levels of free ubiquitin and defective formation of DNA repair compartments at fragile chromosomal sites was previously reported by independent groups (Mailand *et al*, 2007; Chroma *et al*, 2016). Our data identify newly synthesized proteins, including DRiPs, as the main species that upon proteasome inhibition deplete free ubiquitin, thereby compromising DNA damage sensing/repair. Thus, conditions that impair the function of chaperones, proteasomes, and PML pose a major threat to cellular health and survival. Such conditions may include aging and age-related neurodegenerative diseases, such as polyglutamine diseases, where aggregated proteins colocalize with PML-NBs and deplete nuclear ubiquitin (Ben Yehuda *et al*, 2017), and ALS, which is linked to mutations in genes involved in PQC and DNA repair (Gao *et al*, 2017; Walker *et al*, 2017; Chia *et al*, 2018).

# Materials and Methods

### Experimental model

HeLa cells and HeLa Kyoto BAC cell lines were grown at 37°C and 5% $CO_2$ in DMEM high glucose (4.5 g/l) medium supplemented with 2 mM L-glutamine, 100 U/ml penicillin/streptomycin, and 10% fetal bovine serum. The Hela Kyoto BAC cell

lines were kept under selection in geneticin (G-418, Thermo Fisher, 400 μg/ml).

### Transfection, cDNAs, protein extraction, and immunoblotting

Transfections of cDNAs and siRNAs were performed using Lipofectamine 2000 (Life Technologies) following manufacturer's instructions. Experiments were performed 24 and 72 h after transfection of cDNAs or siRNA, respectively.

The cDNAs used were as follows: RPL23a-GFP (RG217630, OriGene); eGFP (Clonetech); GFP-GR50 (Lee *et al*, 2016); and Flag-VCP (Ritson *et al*, 2010); mCherry-PML was generated for this study; GFP-TDP43 (Zhang *et al*, 2009); mCherry-VHL (Mateju *et al*, 2017); NLuc-GFP (Nollen *et al*, 2001); HA-Ubiquitin and Flag-Ubiquitin (den Engelsman *et al*, 2003).

To extract total proteins, cells were lysed in Laemmli sample buffer and homogenized by sonication. Prior to separation by SDS–PAGE, the protein samples were boiled for 3 min at 100°C and reduced with β-mercaptoethanol. Proteins were transferred onto nitrocellulose membranes and analyzed by Western blotting.

### Nucleoli isolation

Nucleoli were isolated from HeLa cells as previously described (Andersen *et al*, 2002) with some variations. Briefly, 5 × 150 mm Petri dishes of confluent HeLa cells were washed three times with cold PBS. Cells were scraped and collected by centrifugation at 218 *g* and 4°C for 5 min. Cells were subsequently resuspended in 5 ml of cold Buffer A (10 mM Hepes pH 7.9, 10 mM KCl, 1.5 mM $MgCl_2$, 0.5 mM DTT, EDTA-free protease inhibitors, Roche) and incubated 15 min on ice. Cells were then dounce homogenized 20 times using a tight pestle and centrifuged at 218 *g* and 4°C for 5 min. The supernatant was collected as cytoplasmic fraction, while the pellet, which contained nuclei, was resuspended with 3 ml of S1 solution (0.25 M sucrose, 10 mM $MgCl_2$, EDTA-free protease inhibitors). Resuspended nuclei were layered over 3 ml of S2 solution (0.35 M sucrose, 0.5 mM $MgCl_2$, EDTA-free protease inhibitors) and centrifuged at 1,430 *g* and 4°C for 5 min. The resulting nuclear pellet was resuspended with 3 ml of S2 solution and sonicated 6 × 10 s bursts at 20% amplitude (Branson Digital Sonifier 450-D). The sonicated sample was layered over 3 ml of S3 solution (0.88 M sucrose, 0.5 mM $MgCl_2$, EDTA-free protease inhibitors) and centrifuged at 3,000 *g* and 4°C for 10 min. The supernatant was collected as nucleoplasmic fraction. The pellet, which contained nucleoli, was resuspended in 0.5 ml of S2 solution, centrifuged at 1,430 *g* and 4°C for 5 min, resuspended in 0.1 ml of S2 solution, and used to assess nucleoli purity and DRiP distribution by Western blotting.

### Immunofluorescence on cultured cells and labeling of nascent peptides with OP-puro

Cells were grown on polylysine-coated glass coverslip. After washing with cold PBS, cells were fixed with 3.7% formaldehyde in PBS for 9 min at room temperature, followed by permeabilization with cold acetone for 5 min at −20°C. Alternatively, cells were fixed with ice-cold methanol for 10 min at −20°C. PBS containing 3% BSA and 0.1% Triton X-100 was used for blocking and incubation with primary and secondary antibodies.

Labeling of newly synthesized proteins was performed by incubating the cells with 25 μM O-propargyl-puromycin (OP-puro) for the indicated time points. OP-puro-labeled peptides were detected by click chemistry as previously described (Ganassi *et al*, 2016).

### Amylo-glo staining of cultured cells

HeLa cells were seeded on non-coated glass coverslips. 24 h later, cells were either left untreated or subjected to stress as described in the main text. Cells were then washed with cold PBS, fixed with 3.7% formaldehyde in PBS for 9 min at room temperature, and permeabilized with PBS containing 0.2% Triton X-100. Cells were stained with Amylo-glo (1X) in 0.9% NaCl for 15 min at room temperature and washed for 5 min with 0.9% NaCl. Cells were immediately analyzed by confocal microscopy.

### High content imaging-based assay

Images were obtained using a Leica SP2 AOBS system (Leica Microsystems) and a 63 × oil immersion lens. PML body number, DRiP foci number, and enrichment for DRiPs and polyUb proteins inside PML bodies were analyzed using the Scan$^R$ Analysis software (Olympus). First, PML bodies were segmented based on PML signal using edge detection algorithm. Following PML body segmentation, we measured the mean fluorescence intensity of the protein of interest in each detected PML body. Additionally, the mean fluorescence intensity of the protein was measured in an area surrounding the PML body. The relative enrichment of the protein in individual PML bodies was calculated as a ratio of the mean fluorescence intensity inside the PML body divided by the mean intensity in the region surrounding the PML body. The values were plotted as column graphs representing the fraction of PML bodies with enrichment > 1.5. Two-tailed *t*-test was performed to compare the enrichment values between two groups.

### Live-cell imaging and Fluorescence recovery after photobleaching

Live-cell imaging was done using the DeltaVision imaging system and SoftWorx 4.1.2 and with the Leica SP8 system. FRAP measurements on PML-GFP HeLa Kyoto cells were performed using a spinning-disk confocal microscope (Olympus IX81), while FRAP measurements on GFP-PSMA7 HeLa Kyoto cells were performed using the Leica SP8 system.

For FRAP analysis on PML-GFP HeLa Kyoto cells, we used a 100× oil immersion objective. A region of approximately 2.02 × 2.02 μm was bleached for 60 ms using a laser intensity of 30% at 405 nm. Recovery was recorded for 120 time points after bleaching (600 s). For FRAP analysis on GFP-PSMA7 HeLa Kyoto cells, we used a 63× oil immersion objective. A region of approximately 2.2–2.5 × 2.2–2.5 μm was bleached for 1 s using a laser intensity of 30% at 405 nm. For FRAP analysis of untreated cells or in cells during the stress recovery in drug-free medium, a laser intensity of 100% for 5 s was used. Recovery was recorded for 600 time points after bleaching (600 s). Analysis of the recovery curves for both PML-GFP and GFP-PSMA7 were carried out with the FIJI/ImageJ.

The flow of the protein within the PML body or the GFP-PSMA7 enriched body was measured by quantifying the recovery of the bleached area at the cost of the unbleached region and using a custom written FIJI/ImageJ routine. The bleached region was corrected for general bleaching during image acquisition. We quantified the molecules that move from the unbleached region to the bleached region, leading to recovery of the bleached region.

Prior to FRAP analysis, we corrected the images for drift using the StackReg plug-in function of the FIJI software suite. The equation used for FRAP analysis is as follows ((Ibleach − Ibackground)/(Ibleach(t0) − Ibackground(to)))/((Itotal-Ibackground)/(Itotal(t0) − Ibackground(to))), where Itotal is the fluorescence intensity of the entire cellular structure, Ibleach represents the fluorescence intensity in the bleach area, and Ibackground the background of the camera offset. FRAP curves were averaged to obtain the mean and standard deviation. Fluorescent density analysis was performed using FIJI/ImageJ and selecting specific region of interest (ROI).

### Colony formation assay

Colony formation assay was performed as previously described (Crowley *et al*, 2016). Briefly, 24 h after seeding, HeLa cells were either left untreated or subjected to proteotoxic stress as described in the main text. Cells were subsequently allowed to recover from stress and form colonies at 37°C, 5% $CO_2$ for 10 days. Then, colonies were fixed with 100% methanol at room temperature for 20 min and stained with 0.5% crystal violet in 25% methanol for 5 min.

### Quantification and statistical analysis

All statistical analyses were performed using one-way ANOVA, followed by Bonferroni–Holm post hoc test for comparisons between three or more groups or Student's *t*-test for comparisons between two groups using Daniel's XL Toolbox.

**Expanded View** for this article is available online.

### Acknowledgements

S.C. acknowledges funding from AriSLA Foundation (Granulopathy and MLOpathy); Cariplo Foundation (Rif. 2014-0703); MAECI (Dissolve_ALS); and MIUR (Departments of excellence 2018–2022; E91I18001480001). S.C. and S.A. are grateful to EU Joint Programme—Neurodegenerative Disease Research (JPND) project. The project is supported through funding organizations under the aegis of JPND (http://www.neurodegenerationresearch.eu/). This project has received funding from the European Union's Horizon 2020 Research and Innovation Programme under grant agreement No 643417. S.A. acknowledges funding from the Max Planck Society, the ERC (no. 725836), and the BMBF (01ED1601A, 031A359A). We thank CIGS and people from the MPI-CBG microscopy facility for technical support. J.G.-B. was supported by an EMBO Long-Term fellowship (ALTF 406-2017). We thank Dr. J.P. Taylor, Dr. H.H. Kampinga, and Dr. W. Boelens for kindly providing us cDNAs.

### Author contributions

LM performed the majority of the experiments reported in this study. JG-B performed nucleoli isolation, FRAP on nucleoli, cloning of mCherry-PML, and assisted LM. JV performed live-cell imaging, STED microscopy, and FRAP with LM. TMF assisted with FRAP analysis and DM performed automated image analysis. IB, ADC, FFM, and TT provided technical support to LM. IP produced

and characterized the HeLa BAC cell lines. SC conceived the project. SC and SA wrote the paper with help of TMF, LM, and JG-B.

## Conflict of interest

The authors declare that they have no conflict of interest.

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
