## [Review Process File · The EMBO Journal]

Defective ribosomal products challenge nuclear function by impairing nuclear condensate dynamics and immobilizing ubiquitin

Laura Mediani, Jordina Guillén-Boixet, Jonathan Vinet, Titus M. Franzmann, Ilaria Bigi, Daniel Mateju, Arianna D. Carrà, Federica F. Morelli, Tatiana Tiago, Ina Poser, Simon Alberti, Serena Carrà.

Review timeline:

Submission date:	10 th December 2018
Additional Editor Correspondence:	4 th February 2019
Additional Author Correspondence:	8 th February 2019
Editorial Decision:	12 th February 2019
Revision received:	17 th April 2019
Editorial Decision:	28 th May 2019
Revision received:	5 th June 2019
Accepted:	7 th June 2019

Editor: Hartmut Vodermaier

Transaction Report:

Additional Editor Correspondence:

4th February 2019

Thank you again for submitting your manuscript on DRiP effects on nuclear architecture and function to The EMBO Journal. After some delay linked to the holiday break, it has now been assessed by three expert reviewers, whose comments are copied below for your information. Although all referees acknowledge the potential significance of your findings, you will see that only referee 1 is currently unconditionally in favor of publication, while referees 2 and 3 raise a number of major conceptual and experimental issues that would need to be addressed prior to further consideration of the study. In particular, the referees are concerned that all analyzed perturbations rely on harsh, broad-spectrum inhibitors (such as OP-puro, CHX, MG132), and further criticize the sole reliance on immunofluorescence as a readout as well as the quality of these data. Since it is not clear how these key problems could be decisively overcome, and whether they might be satisfactorily clarified during a the time frame of regular, single-round revision, I would in this case appreciate hearing from you how you would envision responding to the referees' points should you be given the opportunity to revise this work for The EMBO Journal. Therefore, please carefully consider the attached reports and send back a brief point-by-point response outlining how the referees' comments might be addressed/clarified. These tentative response (parts of which we may choose to share and discuss with referees) would be taken into account when making our final decision on this manuscript. It would be great if you could get back to me with such a response by the end of this week.

REFeree REPORTS

Referee #1 (Report for Author)

In this manuscript by Mediani et al., the authors ask the question how DRiPs (defective ribosomal

products) affect cell function. They show that many DRiPs, in particular short DRiPs, freely diffuse into the nucleus and partition into the nucleolus. There, they lead to protein aggregation and the formation of nucleolar aggresomes or amyloid-bodies, which the authors here show to be the same body and hence call NoABs. The combination of DRiPs with heat or proteotoxic stress results in the accumulation of DRiPs in PML bodies, which transition from a liquid into a solid state, which is characterized by immobilization by PML-associated proteins, but also ubiquitin, proteasomes and other PQC factors. These factors are essential for marking and protection of DNA lesions, and the authors demonstrate that proteostasis processes and DNA repair compete for these factors under stress conditions.

These findings show an interesting mechanistic link between nuclear proteostasis and defective DNA repair compartments. This link occurs via liquid membraneless compartments that change their material properties upon stress via the incorporation of the DRiPs. This manuscript fits into a series of publications that showed a link between stress conditions, aberrant phase transitions and changes in material properties of liquid organelles. Importantly, not only stress granules are affected as also shown in a series of manuscripts on C9orf72 repeat expansions.

This is an important manuscript that produces beautiful data in an elegant series of experiments. In my opinion, it is suitable for publication.

In the light of the above mentioned C9orf72 publications, I would urge the authors to ask the question whether changes in the material properties of PML bodies also have effects on the material properties of other membraneless organelles in the cell, perhaps even including the nuclear pore complex. Is FRAP recovery of the nucleolus or ribosome biogenesis affected?

This is a beautiful paper that I think will have a strong impact on the community. I do not see anything wrong with the conclusions in this manuscript but will leave it to Cell Biology experts to address the suitability of the experimental conditions, controls etc as I am not able to fully judge them.

Referee #2 (Report for Author)

In the presented manuscript Mediani and colleagues analyze the role of aggregation-prone proteins, which are mostly defective ribosomal products (DRiPs) produced by defective translation in the cytoplasm. The authors detected newly synthesized small and unstable DRiPs with puromycin labeling and observed that they accumulate in nucleoli and PML bodies. DRiPs are targeted constitutively to nucleoli, but in contrast, are targeted to PML bodies under stress conditions. If DRiPs are not properly degraded by the proteasome, both nuclear compartments undergo transformation to the solid state. The authors speculate that solid PML bodies reduce the available pool of ubiquitin, which then compromises the formation of ubiquitin-dependent DNA repair foci. This is an interesting manuscript that explores the partially understood effect on genome stability and cell survival of imported defective peptides that accumulate in the nucleus. The experiments are generally well designed and the data are novel. In my opinion, however not all conclusions can be interpreted as easily as they are stated since they are confounded by direct effects of the major inhibitors on both nuclear structures of interest (see below).

Here are my major points of criticism:

The quality of provided microscopic images is not satisfactory. The cells in Figure 2, are in most cases too small, and the enlarged insets are not sufficiently resolved. The authors should show high resolution images of the intranucleolar localization of puromycylated DRiPs. The GFP channels in Figure 2B, C are clearly oversaturated. Consequently, they do not precisely define the subnucleolar compartment(s) in which DRiPs are precisely localized.

In contrast to Figure 1, DRiPs in Figure 2B are localized in numerous non-nucleolar foci throughout the nucleoplasm that do not resemble PML bodies or nuclear speckles. What are these structures?

How frequently are these nuclear foci visible in cells with puromycylated DRiPs?

The signal of Amylo-Glo in Figure 2D, E is very weak and needs to be improved.

Both key inhibitors, cycloheximide and MG132, have prominent effects on nucleoli and PML bodies. Cycloheximide affects intranucleolar organization and ribosomal transcription. The proteasome inhibitor MG132 induces accumulation of proteasome subunits in the nucleolus. It also induces the accumulation of 90S pre-ribosomal particles, changes the dynamics of pre-rRNA

processing, slows down the release of mature rRNA from the nucleolus, and leads to the depletion of mature 18S and 28S rRNAs. This fact raises a concern whether observed phenotypes are secondary effects of global downregulation of the nucleolar function.

The authors claimed that the accumulation of DRiPs in nucleoli does not have an effect on the mobility of the two major GFP-tagged nucleolar proteins. These findings do not exclusively prove that the nucleoli are fully functional. The authors should determine whether or not this accumulation has an effect on ribosomal transcription by the use of qRT-PCR and pre-rRNA processing.

I am not convinced how the use of GFP-tagged heat-sensitive luciferase (NLuc-GFP) as a model substrate contributes to the investigation of the role of PML body as a nuclear compartment used for misfolded proteins. I believe this small luciferase enzyme, which relies on a substrate called furimazine and molecular oxygen to produce bioluminescence, is under normal conditions properly folded. Thus, NLuc GFP-localization in nucleoli and PML body under heat shock conditions, if at some level misfolded, seems to be only correlative and does not mimic DRiPs behavior and dynamics.

The authors observed that the mobility of PML-GFP measured by FRAP was dramatically reduced by >50% in HeLa cells treated with proteasome inhibitor MG132 and OP-Puro in contrast to cells treated with the translation inhibitor cycloheximide. It has been reported that proteins localized in PML body move to the nucleolus upon MG132 treatment. Therefore, observed slowdown effect of PML-GFP mobility could be related to the reorganization of PML body structural integrity and might not be directly linked to the accumulation of DRiPs.

The authors do not indicate which isoform(s) of PML protein they targeted for depletion by siRNA. Since there are many PML isoforms with different mobility, this might be an issue for FRAP comparison and overall mobility.

VCP depletion significantly increases the number of PML bodies even in control cells.

Referee #3 (Report for Author)

Mediani and colleagues provide the characterization of defective ribosomal products - DRiPs - which are the major aggregation-prone proteins within cells. They report that DRiPs passively diffuse to the nucleus, accumulating to nucleoli and PML bodies, where they are processed by chaperons and proteasomal components. If this processing fails, accumulation of DRiPs promotes amyloidogenesis and liquid-solid transition of PML bodies, with the impairment of the all ubiquitin system, leading to impairment of ubiquitin-dependent events within the cells, including the DNA damage and repair processes.

Although the study is potentially of interest, the main conclusions are based on data obtained using a single experimental condition (OP-puro), which is quite harsh and artificial, leading to a general unbalance in protein homeostasis. Moreover, most of the results are based on a single technique, i.e. immunofluorescence, contributing to the weakness of this study. Many additional experiments, using different experimental approaches and conditions, should be done in support of the main findings.

Major points

1. Figure 1. The treatment with OP-puro, which allows to detect the fate of the DRiPs within the cell, blocks the synthesis of all cellular proteins and produces a huge amount of premature terminated proteins, which is highly artefactual.

The authors should rather use a milder treatment of OP-puro, allowing the tagging of the misfolded proteins but in a more 'physio-pathological' context.

Most of the DRiPs seems to go into the nucleus, although it would be expected to see a large part of them in the cytosol, as also suggested by the Western blot in Fig. 1B. It is not clear which is the portion and the size (molecular weight) of nucleolar DRiPs compared to the nuclear ones (not shown) and to the cytosolic fractions.

Please, provide biochemical analysis of the distribution of DRiPs in different cell compartments: cytosol, nucleus, nucleoli.

2. Figure 2-7. General comments for all the figures. Most of the experiments are done in different combinations of very harsh conditions (treatments with CHX, heat shock, MG132, OP-puro), which make difficult to correctly interpret the data.

3. Figure 3. Quantification of the co-localization data are missing and should be provided. DRiPs and PML do not really co-localize but are rather adjacent, which could be in principle an interesting observation. The ubiquitination of misfolded proteins has been largely described, therefore the co-localization of DRiPs and FK1 is not surprising.

Why there is no FK1 signal in the nucleoli, which are full of DRiPs? How are the DRiPs degraded in the nucleoli, considering that both ubiquitin (FK1 staining) and the proteasome (20S) are excluded?

Please, provide biochemical analysis showing the ubiquitination of DRiPs.

4. Figure 5. Again, no data on direct ubiquitination of DRiPs.

5. Figure 6. FRAP technique measures the mobility of a specific protein over time. Using this as readout for the conversion of PML bodies from liquid to solid is a bit limited. Additional proofs should be provided.

6. Figure 7. This figure recapitulates data already present in the literature. Blocking degradation of ubiquitinated proteins by using the proteasome inhibitor MG132 results in the depletion of the pool of free ubiquitin, thereby affecting all ubiquitin-based processes within the cells, including DNA damage response and DNA repair. Indeed, both H2A ubiquitination and recruitment of DDR factors to DNA lesions are largely impaired. See as representative references Mailand et al, Cell 2007; Chroma et al, Oncogene 2017.

Minor points

1. The Result section contains extensive discussion, which should be limited and moved to the Discussion section.
2. In some parts of the Results, the reference to Figures is a bit confusing.

Modena, February 08, 2019

Dear Dr. Hartmut Vodermaier,

Thank you for your letter and the reviewers' comments on our manuscript "DRiPs challenge nuclear function by impairing nuclear condensate dynamics and immobilizing ubiquitin" by Mediani et al.

We were pleased to hear that **referee 1** liked our manuscript and recommended publication: *"This is an important manuscript that produces beautiful data in an elegant series of experiments. In my opinion, it is suitable for publication."*

Referee 2 also found that our study is important: *"This is an interesting manuscript that explores the partially understood effect on genome stability and cell survival of imported defective peptides that accumulate in the nucleus. The experiments are generally well designed and the data are novel."*

However, **referee 2** also raised concerns about the conclusions and wonders *"whether observed phenotypes are secondary effects of global downregulation of the nucleolar function."*

Similarly, **referee 3** found our study *"potentially of interest"*, although raising concerns about the use of *"a single experimental condition (OP-puro)"* and *"a single technique, i.e. immunofluorescence"*.

Inspired by the reviewers' comments, we will add important new data, which strengthen the conclusions and add important weight to the manuscript. In summary, we plan the following major changes:

- 1) We will add new data demonstrating that the observed phenotypes are not secondary effects of global downregulation of the nucleolar function, but rather represent an adaptive stress response to the diffusion into the nucleus of a pool of newly synthesized misfolded proteins (see below point-by-point reply and reply Figure 1).
- 2) Our study does not include a single experimental condition (OP-puro), since compartmentalization of endogenous polyubiquitinated proteins at PML-NBs was observed also in cells exposed to heat shock alone, and not treated with OP-puro (Figure 3B, C).
- 3) In reply to the concern that *"Most of the experiments are done in different combinations of very harsh conditions (treatments with CHX, heat shock, MG132, OP-puro), which make difficult to correctly interpret the data"*, the referees could not suggest alternative experimental approaches, besides the use of *"a milder treatment of OP-puro"*. However, it is important to highlight that the temperature and duration of stress chosen for our experiments are widely used and accepted for the study of the heat shock response (42 °C for 2 hrs). Moreover, we employed a relatively low concentration of MG132 (10 µM), also widely used throughout the scientific literature, and for a short period of time

(4 hrs). Similar combinations of heat shock, puromycin and cycloheximide have been recently used to investigate how heat shock (42 and 43 °C) affects transcription, with intriguing findings implicating active translation and nascent protein ubiquitination (1).

There are two additional important aspects that need to be highlighted. First, when translation was inhibited with cycloheximide, although the cells were exposed to heat shock or MG132, polyubiquitinated (polyUb) protein compartmentalization at PML-NBs and amyloidogenesis within nucleolar subcompartments were not observed, regardless of active RNA synthesis (see reply Figure 1). Second, newly synthesized misfolded proteins that were compartmentalized in nucleolar bodies and PML-NBs upon stress, could be disposed by chaperones and proteasomes during the recovery phase after stress. This strongly supports our interpretation that transient compartmentalization of newly synthesized misfolded proteins in nuclear subcompartments represents an adaptive and protective stress response, and not a passive consequence of cell exposure to harsh conditions.

Finally, we provide new results showing that conversion of the nucleolus into an amyloid-like state occurs also in cells expressing GFP-tagged poly-GR (GFP-GR50) but not exposed to other types of stress. GFP-GR50 is a dipeptide repeat (DPR) that accumulates inside the nucleolus and originates from the GGGGCC (G₄C₂) repeat expansion in a noncoding region of C9ORF72, which represents the most common cause of amyotrophic lateral sclerosis (ALS) and frontotemporal dementia (FTD) (2). We will be happy to include this result in our manuscript.

All other minor issues will be addressed through new experiments or textual revisions.

Like the referees, we believe that understanding how the nucleus copes with proteotoxic stress and the mechanistic link between nuclear proteostasis and defective DNA repair is very important. We hope that, given the overall positive nature of the referees and our ability to address all major concerns raised in the review process, you allow us to submit our revised manuscript.

Sincerely,

Serena Carra

Point-by-point reply to referees' comments:

Referee #1 (Report for Author)

In this manuscript by Mediani et al., the authors ask the question how DRiPs (defective ribosomal products) affect cell function. They show that many DRiPs, in particular short DRiPs, freely diffuse into the nucleus and partition into the nucleolus. There, they lead to protein aggregation and the formation of nucleolar aggresomes or amyloid-bodies, which the authors here show to be the same body and hence call NoABs. The combination of DRiPs with heat or proteotoxic stress results in the accumulation of DRiPs in PML bodies, which transition from a liquid into a solid state, which is characterized by immobilization by PML-associated proteins, but also ubiquitin, proteasomes and other PQC factors. These factors are essential for marking and protection of DNA lesions, and the authors demonstrate that proteostasis processes and DNA repair compete for these factors under stress conditions.

These findings show an interesting mechanistic link between nuclear proteostasis and defective DNA repair compartments. This link occurs via liquid membraneless compartments that change their material properties upon stress via the incorporation of the DRiPs. This manuscript fits into a series of publications that showed a link between stress conditions, aberrant phase transitions and changes in material properties of liquid

organelles. Importantly, not only stress granules are affected as also shown in a series of manuscripts on C9orf72 repeat expansions.

This is an important manuscript that produces beautiful data in an elegant series of experiments. In my opinion, it is suitable for publication.

Reply: We are pleased to hear that referee 1 likes our manuscript and recommends publication.

In the light of the above mentioned C9orf72 publications, I would urge the authors to ask the question whether changes in the material properties of PML bodies also have effects on the material properties of other membraneless organelles in the cell, perhaps even including the nuclear pore complex. Is FRAP recovery of the nucleolus or ribosome biogenesis affected?

Reply: Figure S1D, E show that FRAP recovery of GFP-tagged nucleophosmin 1 (GFP-NPM1), a marker of the granular compartment (GC), and nucleolin (GFP-NCL), a marker of the dense fibrillar compartment (DFC) were very similar in the presence or absence of DRiPs. This observation is in agreement with previous findings reporting the accumulation of proteins inside discrete nucleolar subcompartments, referred to as nucleolar detention center or amyloid-bodies or nucleolar aggresomes upon stress conditions such as heat shock, acidosis, prolonged proteasome inhibition and transcriptional stress (3-6).

Concerning the study of the effect of DRiPs on the material properties of other membraneless organelles in the cell, we previously published that DRiPs promote the conversion of cytoplasmic stress granules into a solid-like state (7). It will certainly be important to study in the future the effect of DRiPs on the morphology and functionality of the nuclear pore complex.

This is a beautiful paper that I think will have a strong impact on the community. I do not see anything wrong with the conclusions in this manuscript but will leave it to Cell Biology experts to address the suitability of the experimental conditions, controls etc as I am not able to fully judge them.

Reply: We thank the referee for acknowledging the importance and novelty of our work.

Referee #2 (Report for Author)

In the presented manuscript Mediani and colleagues analyze the role of aggregation-prone proteins, which are mostly defective ribosomal products (DRiPs) produced by defective translation in the cytoplasm. The authors detected newly synthesized small and unstable DRiPs with puromycin labeling and observed that they accumulate in nucleoli and PML bodies. DRiPs are targeted constitutively to nucleoli, but in contrast, are targeted to PML bodies under stress conditions. If DRiPs are not properly degraded by the proteasome, both nuclear compartments undergo transformation to the solid state. The authors speculate that solid PML bodies reduce the available pool of ubiquitin, which then compromises the formation of ubiquitin-dependent DNA repair foci.

This is an interesting manuscript that explores the partially understood effect on genome stability and cell survival of imported defective peptides that accumulate in the nucleus. The experiments are generally well designed and the data are novel. In my opinion, however not all conclusions can be interpreted as easily as they are stated since they are confounded by direct effects of the major inhibitors on both nuclear structures of interest (see below).

Here are my major points of criticism:

The quality of provided microscopic images is not satisfactory. The cells in Figure 2, are in most cases too small, and the enlarged insets are not sufficiently resolved. The authors should show high resolution images of the intranucleolar localization of puromycylated DRiPs. The GFP channels in Figure 2B, C are clearly

oversaturated. Consequently, they do not precisely define the subnucleolar compartment(s) in which DRiPs are precisely localized.

Reply: We will provide high resolution images of the intranucleolar localization of puromycylated DRiPs and decrease the intensity of the GFP signal in Figure 2B, C as requested.

In contrast to Figure 1, DRiPs in Figure 2B are localized in numerous non-nucleolar foci throughout the nucleoplasm that do not resemble PML bodies or nuclear speckles. What are these structures? How frequently are these nuclear foci visible in cells with puromycylated DRiPs?

Reply: Figure 1 shows the accumulation of DRiPs inside nucleoli in growing, unstressed cells. These DRiPs were cleared from nucleoli upon removal of OP-puro (Figure 1A, recovery). By contrast, when the proteasome was inactivated with MG132, DRiPs remained inside nucleoli for much longer (Figure 1A). Figure 2B shows the subcellular localization of DRiPs in cells treated with OP-puro and concomitantly exposed to heat shock. As discussed in the manuscript *"Upon heat shock, a pool of DRiPs and newly synthesized proteins also accumulated in nuclear foci distinct from nucleoli (Figure 2B). These foci did not colocalize with nuclear speckles, where active proteasomal degradation has been reported (Figure S3)"*. Figure 3A shows that these non-nucleolar foci throughout the nucleoplasm partially colocalize with PML bodies. These DRiP foci perfectly colocalized with polyUb proteins; thus, we used the FK1 antibody, which recognizes polyUb proteins, to quantify the frequency of these non-nucleolar foci; quantification is reported in Figure 3D (heat shock). Importantly, accumulation of endogenous newly synthesized polyUb proteins in non-nucleolar foci that partially colocalize with PML bodies occurs also in cells that are exposed to heat shock without concomitant treatment with OP-puro. In order to compare the number of these foci in cells treated with heat shock alone and heat shock+OP-puro treatment, we decided to quantify the FK1 signal (Figure 3A-D).

The signal of Amylo-Glo in Figure 2D, E is very weak and needs to be improved.

Reply: The Amylo-Glo staining is typically quite weak. We will improve the quality of Figure 2D. Amylo-Glo is not shown in Figure 2E.

Both key inhibitors, cycloheximide and MG132, have prominent effects on nucleoli and PML bodies. Cycloheximide affects intranucleolar organization and ribosomal transcription. The proteasome inhibitor MG132 induces accumulation of proteasome subunits in the nucleolus. It also induces the accumulation of 90S pre-ribosomal particles, changes the dynamics of pre-rRNA processing, slows down the release of mature rRNA from the nucleolus, and leads to the depletion of mature 18S and 28S rRNAs. This fact raises a concern whether observed phenotypes are secondary effects of global downregulation of the nucleolar function.

Reply: The referee is right in pointing out that the proteasome inhibitor MG132 inhibits pre-RNA processing and induces the accumulation of proteasome subunits in the nucleolus. These changes were previously documented by Stavreva et al. (2006) (8). Importantly, the pre-RNA processing changes reported in this manuscript, occurred upon exposure of the cells to a high dose of MG132 (100 μ M) for 2-3 hrs.

As a follow-up of this manuscript, Latonen et al. (2011) reported that lower doses of MG132 (10 μ M), which are sufficient to block proteasome-mediated degradation, did not affect the nascent rRNA synthesis (6). rRNA synthesis inside nucleoli was detected even after 12 hrs of treatment with MG132 (10 μ M). At this time-point, Latonen et al. reported the formation of a dense aggregate within the nucleolus, which they called nucleolar aggresome (NoA). NoA formed independent on nucleolar integrity. Thus, they concluded that *"while MG132 treatment causes nucleolar reorganization, the nucleoli retain RNA pol I transcriptional activity"* (6).

Importantly, throughout our study, we used MG132 (10 μ M) for 4 hrs, a 10-time lower concentration compared to the one reported to affect pre-RNA processing by Stavreva et al. (2006). After treatment of the cells with MG132 10 μ M for 4 hrs, we do not detect quantitative recruitment of proteasomes inside the nucleolus; instead, the 20S proteasomes accumulate with DRiPs and polyUb proteins at PML-NBs (Figure 3G;

asterisk indicates the nucleolus). Thus, our results strongly support the interpretation that after short-term treatment with MG132, compartmentalization of DRiPs at PML-NBs represents a protective mechanism that promotes the efficient clearance of DRiPs by concentrating ubiquitin, 20S proteasomes and chaperones at these specific PQC sites.

Reply Figure 1: amyloidogenesis is not a secondary effect of global downregulation of the nucleolar function

A: Staining of newly synthesized RNA with ethynyl uridine (EU), followed by click-chemistry. Nucleic acid is stained with DAPI. Co-incubation of the cells with EU and actinomycin D (Act.D) for 6 hrs inhibits transcription. A nucleolus full of newly synthesized rRNA is shown.

B: Cells were treated as indicated in the figure. Amyloidogenesis occurs upon heat shock and transcriptional stress (MG132 + Act.D) only when translation is active. Act.D per se does not induce nucleolar amyloidogenesis.

C: qPCR showing the expression levels of HSPA1A mRNA in untreated cells and following exposure to stress. Act.D prevents the induction of HSPA1A mRNA upon heat shock.

D, E: Confocal microscopy showing that HSPA8 and HSPA1A accumulate in the nucleolus of cells upon transcriptional stress (MG132 + Act.D) only when translation is active.

Concerning the possibility that the “observed phenotypes are secondary effects of global downregulation of the nucleolar function”, we performed a series of experiments using actinomycin D (Act.D), an RNA polymerase inhibitor, which potently suppresses RNA synthesis, including rRNA transcription (6). Using

ethynyl uridine (EU) to detect RNA transcription, we confirmed that treatment of the cells with Act.D for 6 hrs blocks transcription (reply Figure 1A).

Architectural rearrangements of the nucleolus and the formation of the amyloid-body was previously reported to occur as an adaptive response to stressor such as acidosis, heat shock and transcriptional stress, which consists of co-treatment of the cells with MG132 and Act.D for 6 hrs (3-5). In agreement with these findings, upon co-treatment of the cells with Act.D and MG132 for 6 hrs we could observe the formation of the amyloid-body, as judged using the Amylo-Glo dye (reply Figure 1B). Next, to understand the implication of DRiPs in this amyloidogenesis process, rather than suppression of transcription and pre-rRNA processing, we co-treated the cells with cycloheximide, MG132 and Act.D. When cells were co-treated with cycloheximide, MG132 and Act.D, amyloid-bodies did not form (reply Figure 1B). We extended these findings to yet another stress condition: heat shock. Amyloid-bodies formed in response to treatment of the cells with heat shock alone or combined with Act.D, but not when translation was inhibited (Figure 1B). Importantly, inhibition of transcription or translation per se did not trigger the formation of amyloid-body (Figure 1B). This result further excludes the possibility that amyloidogenesis inside the nucleolus is a secondary effect of transcription downregulation. As additional control, to confirm transcription inhibition by Act.D, we measured by qPCR the expression level of HSPA1A mRNA, which is induced upon proteotoxic stress. Figure 1C shows that heat shock significantly induces the expression of HSPA1A mRNA; this occurred also upon co-treatment of the cells with cycloheximide. By contrast, Act.D prevented the heat-shock mediated upregulation of HSPA1A mRNA, while not affecting amyloidogenesis.

Finally, in our manuscript we show that the HSPA8 and HSPA1A chaperones are recruited to the amyloid-body along with DRiPs (Figure 2C and S2) (3). In fact, during heat shock, inhibition of translation with cycloheximide completely abrogated DRiP accumulation inside the nucleolus and it also prevented HSPA8 and HSPA1A nucleolar recruitment (Figure 2C and S2). Here, we provide new data showing that HSPA8 and HSPA1A are recruited inside amyloid-bodies that form upon transcriptional stress, but not when translation is concomitantly inhibited (reply Figure E, F). Based on these findings, we conclude that the formation of the amyloid-bodies is not a secondary effect of global downregulation of the nucleolar function, but it is a consequence of the accumulation of DRiPs inside nucleolar subcompartments. Chaperone recruitment into the amyloid-bodies also depends on the accumulation of DRiPs.

We will be happy to include these results in our revised manuscript.

The authors claimed that the accumulation of DRiPs in nucleoli does not have an effect on the mobility of the two major GFP-tagged nucleolar proteins. These findings do not exclusively prove that the nucleoli are fully functional. The authors should determine whether or not this accumulation has an effect on ribosomal transcription by the use of qRT-PCR and pre-rRNA processing.

Reply: We agree with the referee that FRAP analysis of nucleolin and nucleophosmin 1 mobility do not demonstrate that the nucleoli are fully functional. However, using actinomycin D, we find that conversion of nucleoli into an amyloid-like state does not depend on active transcription but, rather, on active translation (reply Figure 1B). Our data are fully in line with previous reports showing that proteins are detained inside nucleolar subcompartments that are distinct from the nucleolus itself (3-6). We will make textual changes to our manuscript to clarify that stress conditions can affect nucleolar function; nevertheless, our data clearly demonstrate that targeting of misfolded proteins to nucleolar subcompartments is not dependent on active (nucleolar) transcription, but on active translation. We will also investigate pre-rRNA processing by qRT-PCR as requested by the referee.

I am not convinced how the use of GFP-tagged heat-sensitive luciferase (NLuc-GFP) as a model substrate contributes to the investigation of the role of PML body as a nuclear compartment used for misfolded proteins. I believe this small luciferase enzyme, which relies on a substrate called furimazine and molecular oxygen to produce bioluminescence, is under normal conditions properly folded. Thus, NLuc GFP-localization in nucleoli and PML body under heat shock conditions, if at some level misfolded, seems to be only correlative and does not mimic DRiPs behavior and dynamics.

Reply: In our studies, to extend our observations from OP-puro labelled DRiPs to other misfolded proteins we used: 1) endogenous polyUb proteins (in cells exposed to heat shock without co-treatment with OP-puro; Figure 3B, C); 2) mCherry-VHL, a misfolded model substrate (Figure 5A-F); 3) NLuc-GFP (Figure S4). These proteins showed similar subcellular distribution upon stress. Since these observations are correlative, we will address this specific comment by textual revision.

The authors observed that the mobility of PML-GFP measured by FRAP was dramatically reduced by >50% in HeLa cells treated with proteasome inhibitor MG132 and OP-Puro in contrast to cells treated with the translation inhibitor cycloheximide. It has been reported that proteins localized in PML body move to the nucleolus upon MG132 treatment. Therefore, observed slowdown effect of PML-GFP mobility could be related to the reorganization of PML body structural integrity and might not be directly linked to the accumulation of DRiPs.

A: PML-GFP HeLa Kyoto cells expressing mCherry-VHL. 24 h posttransfection, cells were left untreated or treated with MG132 (10 μ M) and OP-puro (25 μ M) for 4 h. Where indicated (OP-puro + MG132 + CHX), translation was inhibited during stress with cycloheximide (CHX; 50 μ g/ml). The subcellular distribution of PML-GFP and mCherry-VHL was studied by live-cell confocal imaging. Representative pictures are shown. Scale bars: 5 μ m.

Reply: To monitor the presence of misfolded proteins inside PML-NBs and assess their direct impact on PML-GFP dynamics, we performed live-cell imaging in cells expressing mCherry-VHL, used as misfolded model substrate (Figure 6A). We will include in this figure a panel showing that upon concomitant treatment of PML-GFP cells with MG132, OP-puro and cycloheximide, mCherry-VHL does not accumulate at PML-NBs (see reply Figure 2); this correlates with FRAP analysis of PML-GFP mobility (compare Figure 6C and D) and strongly supports our interpretation that newly synthesized misfolded proteins that accumulate at PML-NBs affect their mobility.

The authors do not indicate which isoform(s) of PML protein they targeted for depletion by siRNA. Since there are many PML isoforms with different mobility, this might be an issue for FRAP comparison and overall mobility.

Reply: For our studies, we used a Dharmacon SMARTpool: ON-TARGETplus PML siRNA (L-006547-00-0005), which consists of four individual sequences targeting the following PML transcripts (<https://dharmacon.horizondiscovery.com/rnai/sirna/on-targetplus/on-targetplus-sirna-reagents-human/?sourceid=entrezgene/5371>):

- NM_033244 Homo sapiens promyelocytic leukemia (PML), transcript variant 5, mRNA
- NM_033247 Homo sapiens promyelocytic leukemia (PML), transcript variant 8, mRNA
- NM_033249 Homo sapiens promyelocytic leukemia (PML), transcript variant 10, mRNA
- NM_033250 Homo sapiens promyelocytic leukemia (PML), transcript variant 11, mRNA
- NM_033246 Homo sapiens promyelocytic leukemia (PML), transcript variant 7, mRNA
- NM_033240 Homo sapiens promyelocytic leukemia (PML), transcript variant 2, mRNA
- NM_033238 Homo sapiens promyelocytic leukemia (PML), transcript variant 1, mRNA
- NM_002675 Homo sapiens promyelocytic leukemia (PML), transcript variant 6, mRNA
- NM_033239 Homo sapiens promyelocytic leukemia (PML), transcript variant 9, mRNA

Concerning protein detection by immunoblotting, we used the PML antibody (abcam 179466; observed band size: 50-110 kDa). This antibody was raised using a recombinant fragment within the human PML protein from amino acid 150-400, which are common to all PML isoforms (9). According to the manufacturer, the observed band size are from 50-110 kDa (https://www.abcam.com/PML-Protein-antibody-EPR16792-ab179466.html#description_images_1). We now show that PML siRNA treatment leads to a global reduction in the levels of PML. We can include this immunoblotting in our revised manuscript.

VCP depletion significantly increases the number of PML bodies even in control cells.

Reply: VCP is required to degrade DRiPs (10). It is thus plausible that VCP depletion may lead to the accumulation of DRiPs at PML bodies, whose number slightly increases already in resting cells as pointed-out by the referee.

Referee #3 (Report for Author)

Mediani and colleagues provide the characterization of defective ribosomal products - DRiPs - which are the major aggregation-prone proteins within cells. They report that DRiPs passively diffuse to the nucleus, accumulating to nucleoli and PML bodies, where they are processed by chaperons and proteasomal components. If this processing fails, accumulation of DRiPs promotes amyloidogenesis and liquid-solid transition of PML bodies, with the impairment of the all ubiquitin system, leading to impairment of ubiquitin-dependent events within the cells, including the DNA damage and repair processes.

Although the study is potentially of interest, the main conclusions are based on data obtained using a single experimental condition (OP-puro), which is quite harsh and artificial, leading to a general unbalance in protein homeostasis. Moreover, most of the results are based on a single technique, i.e. immunofluorescence, contributing to the weakness of this study. Many additional experiments, using different experimental approaches and conditions, should be done in support of the main findings.

Major points

1. Figure 1. The treatment with OP-puro, which allows to detect the fate of the DRiPs within the cell, blocks the synthesis of all cellular proteins and produces a huge amount of premature terminated proteins, which is highly artefactual. The authors should rather use a milder treatment of OP-puro, allowing the tagging of the misfolded proteins but in a more 'physio-pathological' context.

Reply Figure 3: DRiPs accumulate inside the nucleolus within few minutes

A: DRiP labelling in HeLa cells that were treated with OP-puro for 1, 5, 10 or 15 min. Nucleoli were visualized using a fibrillarin specific antibody. Note that DRiPs can be detected inside nucleoli already after 5 min of treatment.

Reply: Reducing the concentration of OP-puro used for our studies will increase the signal/noise ratio.

We instead incubated the cells for shorter times and we could detect DRiPs inside nucleoli already after 5 min of treatment, with a progressive accumulation in time. The time-course accumulation of DRiPs inside the nucleolus, visualized using fibrillarin, is now included in reply Figure 3 for the referee.

Importantly, as previously mentioned, to avoid misinterpretation due to the treatment of cells with OP-puro, we observed nucleolar amyloidogenesis and accumulation of newly synthesized polyUb proteins in cells that were only subjected to heat shock (without addition of OP-puro; Figure 3B-D).

Most of the DRiPs seems to go into the nucleus, although it would be expected to see a large part of them in the cytosol, as also suggested by the Western blot in Fig. 1B. It is not clear which is the portion and the size (molecular weight) of nucleolar DRiPs compared to the nuclear ones (not shown) and to the cytosolic fractions. Please, provide biochemical analysis of the distribution of DRiPs in different cell compartments: cytosol, nucleus, nucleoli.

Reply: The nucleoli isolation protocol used for our studies does not allow to separate nucleoplasmic proteins from cytoplasmic proteins (11). Using this protocol, we obtain a total fraction, a cytoplasmic and nucleoplasmic (CN) fraction and a nucleolar fraction. We provide for the referee an immunoblot showing the distribution of DRiPs in the total fraction, the CN fraction and the nucleolar fraction. As pointed out by the referee, a large part of DRiPs is found in the CN fraction.

As requested, we will try a different nucleoli isolation protocol that will allow us to obtain separated cytosolic and nucleoplasmic fractions.

2. Figure 2-7. General comments for all the figures. Most of the experiments are done in different

combinations of very harsh conditions (treatments with CHX, heat shock, MG132, OP-puro), which make difficult to correctly interpret the data.

Reply: As previously mentioned, we observed recruitment of endogenous newly synthesized polyUb proteins in the nucleus of cells that were only subjected to heat shock at 42°C for 2 hrs (without addition of OP-puro; Figure 3B-D), which is widely used and accepted to study cell response to temperature upshift. In addition, we use relatively low concentration of the proteasome inhibitor MG132 (10 μM) and for short time-point (4 hrs) to avoid misinterpretation due to extreme and prolonged stress conditions. Finally, similar combinations of heat shock, puromycin and cycloheximide have been recently used to investigate how heat shock (42 and 43 °C) affects transcription, with intriguing findings implicating active translation and nascent protein ubiquitination (1).

Importantly, the morphological and dynamic changes observed at the level of nucleoli and PML-NBs were not observed in cells that were exposed to stress but concomitantly treated with cycloheximide to stop translation. This is an important finding because it demonstrates that the rearrangements of nucleolar subcompartments and PML-NBs that we describe here are not a consequence of the harsh stress condition per se but, instead, they depend on the accumulation of newly synthesized misfolded proteins in the nucleus upon stress. Most importantly, when cells were allowed to recover from stress, newly synthesized misfolded proteins were cleared by chaperones and proteasomes from nucleolar amyloid-bodies and PML-NBs. Thus, we propose that compartmentalization of DRiPs represents a protective mechanism that avoids the promiscuous interaction of DRiPs with essential nuclear components and promotes the efficient clearance of DRiPs by concentrating ubiquitin, 20S proteasomes and chaperones at specific PQC sites.

Reply Figure 4: Nucleoli accumulating GFP-GR50 acquire an amyloid-like state in absence of any external stress

HeLa cells were transiently transfected with a cDNA encoding GFP-GR50 (kindly provided by prof. Taylor). 24 hrs post-transfection cells were stained with amylo-glo to visualize amyloid-like structures.

Finally, to further support our interpretation that misfolded proteins that accumulate inside nucleoli cause their conversion into an amyloid-like state, we tested the effect of GFP-tagged poly-GR (GFP-GR50). GFP-GR50 is a dipeptide repeat (DPR) that accumulates inside the nucleolus and originates from the GGGGCC (G₄C₂) repeat expansion in a noncoding region of C9ORF72, which represents the most common cause of amyotrophic lateral sclerosis (ALS) and frontotemporal dementia (FTD) (2). Nucleoli that accumulate GFP-GR50, in absence of any other type of stress, convert into an amyloid-like state (reply Figure 4). Conversion of nucleoli

into an amyloid-like state in cells expressing GFP-GR50 could contribute to toxicity together with changes in nucleolar functions and nucleocytoplasmic transport, as previously documented (2, 12). We will be happy to include this result in our manuscript.

3. Figure 3. Quantification of the co-localization data are missing and should be provided. DRiPs and PML do not really co-localize but are rather adjacent, which could be in principle an interesting observation. The ubiquitination of misfolded proteins has been largely described, therefore the co-localization of DRiPs and FK1 is not surprising.

Reply: The number of DRiPs foci, their enrichment for polyUb proteins (FK1), as well as the number of PML-NBs enriched for polyUb proteins have been quantified using a high-content automated imaging assay (ScanR, Olympus). High-content automated quantifications are reported in Figure 4B, C, Figure 5C, E and Fig.

S6D. In addition, we quantified the % of cells with FK1 nuclear foci upon stress; these quantifications are reported in Figure 3D, F and Figure 5B, D.

Why there is no FK1 signal in the nucleoli, which are full of DRiPs? How are the DRiPs degraded in the nucleoli, considering that both ubiquitin (FK1 staining) and the proteasome (20S) are excluded? Please, provide biochemical analysis showing the ubiquitination of DRiPs.

Reply: As previously published, proteasome-mediated degradation does not occur inside the nucleolus; instead, it was proposed to occur in discrete foci within the nucleoplasm, including PML-NBs (13-15). Our data are in line with these findings and suggest that misfolded proteins are extracted from the nucleolar subcompartments for disposal. In agreement with this model, we find that chaperones are recruited inside the nucleolus during stress, when amyloidogenesis occurs and are required for the clearance of misfolded proteins from the amyloid-bodies during the recovery phase after stress (our manuscript) (5). The finding that ubiquitin and 20S proteasomes concentrations are high in proximity to PML-NBs strongly supports our interpretation that degradation of polyUb proteins does not occur inside the nucleoli.

Concerning the biochemical analysis showing the ubiquitination of DRiPs, see below.

4. Figure 5. Again, no data on direct ubiquitination of DRiPs.

Reply: It has been previously published that DRiPs are degraded by the proteasomes (16). In addition, in a series of elegant experiments, Wang et al. recently demonstrated that polysome-associated puromycylated nascent polypeptides are ubiquitinated in cells (17). Using fluorescently labeled puromycin, the authors estimated that circa 15% of the total nascent polypeptides are co-translationally ubiquitinated. This percentage further increases upon stress conditions that induce protein misfolding (17). Finally, the authors found that puromycylated nascent chains mainly contain the K48-linked polyubiquitin chains, which is consistent with the targeting of these products to proteasome for degradation. Of note, the FK1 antibody that we use in our microscopy studies recognizes K48-linked poly ubiquitinated and mono ubiquitinated proteins but not free ubiquitin.

We could perform biochemical experiments to demonstrate that DRiPs (puromycylated nascent chains) are ubiquitinated. However, since biochemical purification of PML-NBs is unfeasible, it is important to note that these experiments will not add any information concerning the ubiquitination of the fraction of DRiPs that is recruited at PML-NBs. By contrast, colocalization studies, as done throughout our manuscript, allowed us to show that only the pools of DRiPs that accumulate adjacent to PML-NBs, but not those that accumulate inside nucleoli, colocalize with polyUb conjugates. Next, the experiment shown in Figure 4E clearly shows that *“accumulation of polyUb at PML-NBs depended on active translation and conversely, inhibition of translation prevented the sequestration of DRiPs and polyUb conjugates at PML-NBs (Figure 3A), suggesting that DRiPs become ubiquitinated.”*

We could verify the ubiquitination profile of DRiPs from purified nucleolar extracts, but we wonder how informative would be this experiment considering the microscopy data and the fact that biochemical purification of PML-NBs is unfeasible.

5. Figure 6. FRAP technique measures the mobility of a specific protein over time. Using this as readout for the conversion of PML bodies from liquid to solid is a bit limited. Additional proofs should be provided.

Reply: Changes in the number of PML-NBs in response to stress, including heat shock, DNA damage and viral infection have been extensively documented (18). Thus, besides FRAP analysis to study the mobility of PML-GFP molecules, we also quantified the changes in the number of PML-NBs upon stress and during the recovery time after stress. PML-NB number was quantified using a high-content automated imaging assay (ScanR, Olympus) (Figure S6E, F; additional quantifications of PML-NB number are reported in Figures 4 and 5).

Liquid-like PML-NBs rapidly dissolved during the recovery time after stress; consequently, their total number decreased. By contrast, when proteasome or chaperone function were inhibited during the recovery time,

not only PML-NBs became solid (as measured by FRAP), but their number did not decrease as efficiently as in control cells (Figure S6E, F). This type of analysis is widely used to study the conversion of other types of membraneless organelles, such as stress granules, into an aggregated-like state (7, 19, 20). Our results were then further validated by live-cell imaging studies in cells expressing PML-GFP (Video S1-S5). Thus, the persistence in time of PML-NBs has been used as an additional method to evaluate their solidification. We will highlight this analysis with textual revision.

6. Figure 7. This figure recapitulates data already present in the literature. Blocking degradation of ubiquitinated proteins by using the proteasome inhibitor MG132 results in the depletion of the pool of free ubiquitin, thereby affecting all ubiquitin-based processes within the cells, including DNA damage response and DNA repair. Indeed, both H2A ubiquitination and recruitment of DDR factors to DNA lesions are largely impaired. See as representative references Mailand et al, Cell 2007; Chroma et al, Oncogene 2017.

Reply: We agree with the referee that previous groups published that the local recruitment of DDR factors to DNA damage sites is impaired by depletion of nuclear ubiquitin. However, the majority of these studies (included the ones reported by the referee) were performed in cells that were exposed to DNA damaging agents that cause DNA double-strand breaks such as e.g. ionizing irradiation. Here, we do not treat the cells with DNA damaging agents. In line with these findings, we demonstrate that also the recruitment of repair factors at spontaneous DNA lesions that form in growing cells depends on the availability of ubiquitin; however, we provide strong evidence for a competition between nuclear proteostasis and DNA damage repair. The novelty of our results stands in the finding that a fraction of DRiPs is continuously targeted to the nucleus; if not properly handled, DRiPs limit the available pool of ubiquitin and pose a major threat to cells, which then can no longer maintain a healthy genome.

Minor points

1. The Result section contains extensive discussion, which should be limited and moved to the Discussion section.

Reply: We will modify the text as suggested.

2. In some parts of the Results, the reference to Figures is a bit confusing.

Reply: We will ameliorate the reference to Figures.

REFERENCES

1. Aprile-Garcia F, Tomar P, Hummel B, Khavaran A, Sawarkar R. Nascent-protein ubiquitination is required for heat shock-induced gene downregulation in human cells. *Nat Struct Mol Biol.* 2019;26(2):137-46.
2. Freibaum BD, Lu Y, Lopez-Gonzalez R, Kim NC, Almeida S, Lee KH, et al. GGGGCC repeat expansion in C9orf72 compromises nucleocytoplasmic transport. *Nature.* 2015;525(7567):129-33.
3. Jacob MD, Audas TE, Uniacke J, Trinkle-Mulcahy L, Lee S. Environmental cues induce a long noncoding RNA-dependent remodeling of the nucleolus. *Mol Biol Cell.* 2013;24(18):2943-53.
4. Audas TE, Jacob MD, Lee S. Immobilization of proteins in the nucleolus by ribosomal intergenic spacer noncoding RNA. *Mol Cell.* 2012;45(2):147-57.
5. Audas TE, Audas DE, Jacob MD, Ho JJ, Khacho M, Wang M, et al. Adaptation to Stressors by Systemic Protein Amyloidogenesis. *Dev Cell.* 2016;39(2):155-68.
6. Latonen L, Moore HM, Bai B, Jaamaa S, Laiho M. Proteasome inhibitors induce nucleolar aggregation of proteasome target proteins and polyadenylated RNA by altering ubiquitin availability. *Oncogene.* 2011;30(7):790-805.
7. Ganassi M, Mateju D, Bigi I, Mediani L, Poser I, Lee HO, et al. A Surveillance Function of the HSPB8-BAG3-HSP70 Chaperone Complex Ensures Stress Granule Integrity and Dynamism. *Mol Cell.* 2016;63(5):796-810.
8. Stavreva DA, Kawasaki M, Dundr M, Koberna K, Muller WG, Tsujimura-Takahashi T, et al. Potential roles for ubiquitin and the proteasome during ribosome biogenesis. *Mol Cell Biol.* 2006;26(13):5131-45.
9. Nisole S, Maroui MA, Mascle XH, Aubry M, Chelbi-Alix MK. Differential Roles of PML Isoforms. *Front Oncol.* 2013;3:125.
10. Verma R, Oania RS, Kolawa NJ, Deshaies RJ. Cdc48/p97 promotes degradation of aberrant nascent polypeptides bound to the ribosome. *eLife.* 2013;2:e00308.
11. Li ZF, Lam YW. A new rapid method for isolating nucleoli. *Methods Mol Biol.* 2015;1228:35-42.
12. Kwon I, Xiang S, Kato M, Wu L, Theodoropoulos P, Wang T, et al. Poly-dipeptides encoded by the C9orf72 repeats bind nucleoli, impede RNA biogenesis, and kill cells. *Science.* 2014;345(6201):1139-45.
13. Scharf A, Rockel TD, von Mikecz A. Localization of proteasomes and proteasomal proteolysis in the mammalian interphase cell nucleus by systematic application of immunocytochemistry. *Histochem Cell Biol.* 2007;127(6):591-601.
14. Rockel TD, Stuhlmann D, von Mikecz A. Proteasomes degrade proteins in focal subdomains of the human cell nucleus. *J Cell Sci.* 2005;118(Pt 22):5231-42.
15. Guo L, Giasson BI, Glavis-Bloom A, Brewer MD, Shorter J, Gitler AD, et al. A cellular system that degrades misfolded proteins and protects against neurodegeneration. *Mol Cell.* 2014;55(1):15-30.
16. Schubert U, Anton LC, Gibbs J, Norbury CC, Yewdell JW, Bannink JR. Rapid degradation of a large fraction of newly synthesized proteins by proteasomes. *Nature.* 2000;404(6779):770-4.
17. Wang F, Durfee LA, Huibregtse JM. A cotranslational ubiquitination pathway for quality control of misfolded proteins. *Mol Cell.* 2013;50(3):368-78.
18. Lallemand-Breitenbach V, de The H. PML nuclear bodies: from architecture to function. *Curr Opin Cell Biol.* 2018;52:154-61.
19. Mackenzie IR, Nicholson AM, Sarkar M, Messing J, Purice MD, Pottier C, et al. TIA1 Mutations in Amyotrophic Lateral Sclerosis and Frontotemporal Dementia Promote Phase Separation and Alter Stress Granule Dynamics. *Neuron.* 2017;95(4):808-16 e9.
20. Lee KH, Zhang P, Kim HJ, Mitrea DM, Sarkar M, Freibaum BD, et al. C9orf72 Dipeptide Repeats Impair the Assembly, Dynamics, and Function of Membrane-Less Organelles. *Cell.* 2016;167(3):774-88 e17.

Thank you for response letter to our referees' comments and proposal for revising your manuscript to address the concerns raised in them. I have now had a chance to carefully consider your letter, and I was glad to see that you seem to be in a good position to respond to the most pertinent issues through additional data and clarifications. While I obviously cannot predict the outcome of eventual re-assessment, which will also depend on convincing the critical referees, I would in this light nevertheless be happy to provide you an opportunity to revise the manuscript for The EMBO Journal, along the lines suggested in your tentative response letter. In particular, it shall be important to incorporate the data already provided in your letter into the revised manuscript, and to conduct also some complimentary biochemical analyses related to the ubiquitination status of DRiPs.

Manuscript EMBOJ-2018-101341: “DRiPs challenge nuclear function by impairing nuclear condensate dynamics and immobilizing ubiquitin” by Mediani et al.

Point-by-point reply to referees' comments:

Referee #1 (Report for Author)

In this manuscript by Mediani et al., the authors ask the question how DRiPs (defective ribosomal products) affect cell function. They show that many DRiPs, in particular short DRiPs, freely diffuse into the nucleus and partition into the nucleolus. There, they lead to protein aggregation and the formation of nucleolar aggresomes or amyloid-bodies, which the authors here show to be the same body and hence call NoABs. The combination of DRiPs with heat or proteotoxic stress results in the accumulation of DRiPs in PML bodies, which transition from a liquid into a solid state, which is characterized by immobilization by PML-associated proteins, but also ubiquitin, proteasomes and other PQC factors. These factors are essential for marking and protection of DNA lesions, and the authors demonstrate that proteostasis processes and DNA repair compete for these factors under stress conditions.

These findings show an interesting mechanistic link between nuclear proteostasis and defective DNA repair compartments. This link occurs via liquid membraneless compartments that change their material properties upon stress via the incorporation of the DRiPs. This manuscript fits into a series of publications that showed a link between stress conditions, aberrant phase transitions and changes in material properties of liquid organelles. Importantly, not only stress granules are affected as also shown in a series of manuscripts on C9orf72 repeat expansions.

This is an important manuscript that produces beautiful data in an elegant series of experiments. In my opinion, it is suitable for publication.

Reply: We are pleased to hear that referee 1 likes our manuscript and recommends publication.

In the light of the above mentioned C9orf72 publications, I would urge the authors to ask the question whether changes in the material properties of PML bodies also have effects on the material properties of other membraneless organelles in the cell, perhaps even including the nuclear pore complex. Is FRAP recovery of the nucleolus or ribosome biogenesis affected?

Reply: Figure EV1E, F shows that FRAP recovery of GFP-tagged nucleophosmin 1 (GFP-NPM1), a marker of the granular compartment, and nucleolin (GFP-NCL), a marker of the dense fibrillar compartment are very similar in the presence or absence of DRiPs. This observation is in agreement with previous findings reporting the accumulation of proteins inside discrete nucleolar subcompartments, referred to as amyloid-bodies or nucleolar aggresomes, upon stress conditions such as heat shock, acidosis, prolonged proteasome inhibition and transcriptional stress (1-4).

Concerning ribosome biogenesis, we measured the expression levels of precursor and mature ribosomal RNAs (rRNAs) by RT-qPCR. We find that treatment of the cells with puromycin alone or combined with MG132 for 4 hrs leads to a mild accumulation of precursor and mature 18S rRNAs, as well as 45S rRNA (see revised Figure EV1D). In addition, since referee#2 asked “*whether observed phenotypes are secondary effects of global downregulation of the nucleolar function*”, we also performed additional experiments using transcriptional stress (actinomycin D and MG132), which was previously reported to induce A-body formation (3). We now show that A-body formation requires active translation and depends on the nucleolar accumulation of DRiPs, regardless of nucleolar transcription and rRNA processing (see revised Figure 2F, Appendix Figure S1 and point-by-point reply to referee #2, below). Combined, these data clearly demonstrate that amyloidogenesis within nucleolar subcompartments is a consequence of DRiP accumulation.

Concerning the study of the effect of DRiPs on the material properties of other membraneless organelles in the cell, we previously published that DRiPs promote the conversion of cytoplasmic stress granules into a

solid-like state (5). It will certainly be important to study in the future the effect of DRiPs on the morphology and functionality of the nuclear pore complex.

This is a beautiful paper that I think will have a strong impact on the community. I do not see anything wrong with the conclusions in this manuscript but will leave it to Cell Biology experts to address the suitability of the experimental conditions, controls etc as I am not able to fully judge them.

Reply: We thank the referee for acknowledging the importance and novelty of our work.

Referee #2 (Report for Author)

In the presented manuscript Mediani and colleagues analyze the role of aggregation-prone proteins, which are mostly defective ribosomal products (DRiPs) produced by defective translation in the cytoplasm. The authors detected newly synthesized small and unstable DRiPs with puromycin labeling and observed that they accumulate in nucleoli and PML bodies. DRiPs are targeted constitutively to nucleoli, but in contrast, are targeted to PML bodies under stress conditions. If DRiPs are not properly degraded by the proteasome, both nuclear compartments undergo transformation to the solid state. The authors speculate that solid PML bodies reduce the available pool of ubiquitin, which then compromises the formation of ubiquitin-dependent DNA repair foci.

This is an interesting manuscript that explores the partially understood effect on genome stability and cell survival of imported defective peptides that accumulate in the nucleus. The experiments are generally well designed and the data are novel. In my opinion, however not all conclusions can be interpreted as easily as they are stated since they are confounded by direct effects of the major inhibitors on both nuclear structures of interest (see below).

Here are my major points of criticism:

The quality of provided microscopic images is not satisfactory. The cells in Figure 2, are in most cases too small, and the enlarged insets are not sufficiently resolved. The authors should show high resolution images of the intranucleolar localization of puromycylated DRiPs. The GFP channels in Figure 2B, C are clearly oversaturated. Consequently, they do not precisely define the subnucleolar compartment(s) in which DRiPs are precisely localized.

Reply: All the images shown in Figure 2 (with the exception of the DRiP staining shown in panel D) have been replaced with high quality images. We also provide higher resolution images of the intranucleolar localization of DRiPs and we decreased the intensity of the GFP signal as requested in panels B and C. In addition, we changed the pictures shown in Figure EV2D to avoid saturation of the GFP-NLC green signal and to better show that, during heat shock, HSPA1A accumulates within nucleolar subcompartments. Finally, we provide STED super-resolution microscopy images showing the nucleolar distribution of DRiPs and GFP-NLC, which often overlap but do not always colocalize (revised Figure 1B).

In contrast to Figure 1, DRiPs in Figure 2B are localized in numerous non-nucleolar foci throughout the nucleoplasm that do not resemble PML bodies or nuclear speckles. What are these structures? How frequently are these nuclear foci visible in cells with puromycylated DRiPs?

Reply: Figure 1 shows the accumulation of DRiPs inside nucleoli in growing, unstressed cells. These DRiPs are cleared from nucleoli by proteasomes upon removal of OP-puro (Figure 1A, B, recovery). Figure 2B on the other hand shows the subcellular localization of DRiPs in cells treated with OP-puro and concomitantly exposed to heat shock. Our manuscript contains the following sentence to make the reader aware of these assemblies *“Upon heat shock, a pool of DRiPs and newly synthesized proteins also accumulated in nuclear foci distinct from nucleoli (Figure 2B). These foci did not colocalize with nuclear speckles, where active proteasomal degradation has been reported (Figure EV3A)”*. Figure 3A shows that these non-nucleolar foci colocalize with PML bodies and with polyUb proteins. We thus used the FK1 antibody, which recognizes

polyUb proteins, to quantify the frequency of these non-nucleolar foci; quantification is reported in Figure 3D (OP-puro + heat shock).

The signal of Amylo-Glo in Figure 2D, E is very weak and needs to be improved.

Reply: As requested, we improved the quality of Amylo-Glo staining throughout Figure 2. We also included new data in panels A and F using the RNA polymerase inhibitor Actinomycin D (Act. D; see below).

Both key inhibitors, cycloheximide and MG132, have prominent effects on nucleoli and PML bodies. Cycloheximide affects intranucleolar organization and ribosomal transcription. The proteasome inhibitor MG132 induces accumulation of proteasome subunits in the nucleolus. It also induces the accumulation of 90S pre-ribosomal particles, changes the dynamics of pre-rRNA processing, slows down the release of mature rRNA from the nucleolus, and leads to the depletion of mature 18S and 28S rRNAs. This fact raises a concern whether observed phenotypes are secondary effects of global downregulation of the nucleolar function. The authors claimed that the accumulation of DRiPs in nucleoli does not have an effect on the mobility of the two major GFP-tagged nucleolar proteins. These findings do not exclusively prove that the nucleoli are fully functional. The authors should determine whether or not this accumulation has an effect on ribosomal transcription by the use of qRT-PCR and pre-rRNA processing.

Reply: The referee is correct in pointing out that the proteasome inhibitor MG132 inhibits pre-rRNA processing and induces the accumulation of proteasome subunits in the nucleolus. These changes were previously documented by Stavreva et al. (2006) (6). The pre-rRNA processing changes reported in Stavreva et al. occurred upon exposure of the cells to a high dose of MG132 (100 μ M) for 2-3 hrs. As a follow-up of this manuscript, Latonen et al. (2011) reported that lower doses of MG132 (10 μ M), which are sufficient to block proteasome-mediated degradation, did not affect nascent rRNA synthesis (4). rRNA synthesis inside nucleoli was detected even after 12 hrs of treatment with 10 μ M MG132 (4). Throughout our study, we used MG132 (10 μ M) for 4 hrs, a 10-times lower concentration compared to the one reported to affect pre-rRNA processing by Stavreva et al. (2006).

To specifically test the possibility that the “observed phenotypes are secondary effects of global downregulation of the nucleolar function”, we evaluated by RT-qPCR the levels of precursor (45S, 18S 5'J, 5.8S 5'J RNAs) and mature (18S, 28S and 5.8S RNAs) ribosomal RNAs in cells treated for 4 hrs with puromycin alone or combined with MG132 (10 μ M). As shown in revised Figure EV1D, DRiPs and MG132 for 4 hrs did not deplete rRNAs. Instead, we observed a mild accumulation of precursor and mature 18S rRNAs, as well as 45S rRNA (revised Figure EV1D). In our analysis, we included as control the RNA polymerase inhibitor Act.D, which potently suppresses RNA synthesis, including rRNA transcription (4). As expected, Act. D significantly decreased the synthesis of precursor rRNAs (revised Figure EV1D). We further confirmed that treatment of the cells with Act. D blocks transcription using ethynyl uridine (EU) to detect newly synthesized RNA (revised Figure EV2A). Of note, global downregulation of nucleolar function with Act. D did not induce A-body formation (revised Figure 2A, F). This suggests that A-body formation may occur independently on nucleolar transcription. In line with this interpretation, A-body formation was previously reported to occur also upon transcriptional stress, which consists of co-treatment of the cells with MG132 and Act. D for 6 hrs (3). We confirmed that transcriptional stress induced A-body formation, as judged using the Amylo-Glo dye (revised Figure 2F). In parallel, by RT-qPCR we now show that transcriptional stress profoundly affects rRNA transcription (revised Appendix Figure S1). Importantly, A-body formation during transcriptional stress occurred only when translation was active and DRiPs accumulated in nucleolar subcompartments (revised Figure 2F), demonstrating that nucleolar amyloidogenesis depends on DRiPs.

Additionally, we extended these findings to another stress condition: heat shock. Revised Figure EV2C shows that heat shock significantly induces the expression of HSPA1A mRNA; this occurred also upon co-treatment of the cells with cycloheximide. By contrast, Act. D prevented the heat-shock mediated upregulation of HSPA1A mRNA. Concerning A-body formation, it only occurred in response to heat shock alone or combined with Act. D when translation was active (revised Figure 2A). Thus, also upon heat shock, nucleolar amyloidogenesis occurred in a translation-dependent and transcription-independent manner.

Finally, in our manuscript we showed that the HSPA8 and HSPA1A chaperones are recruited to the A-body along with DRiPs (Figure 2C and EV2D) (1). We now show that HSPA8 and HSPA1A are recruited inside A-bodies only when translation is active, regardless of transcription inhibition (revised Figure EV2, new panels E and F). Collectively, our findings demonstrate that A-body formation is not a secondary effect of global downregulation of nucleolar function, but it is a consequence of the accumulation of DRiPs inside nucleolar subcompartments. Chaperone recruitment into A-bodies depends on the accumulation of DRiPs.

I am not convinced how the use of GFP-tagged heat-sensitive luciferase (NLuc-GFP) as a model substrate contributes to the investigation of the role of PML body as a nuclear compartment used for misfolded proteins. I believe this small luciferase enzyme, which relies on a substrate called furimazine and molecular oxygen to produce bioluminescence, is under normal conditions properly folded. Thus, NLuc GFP-localization in nucleoli and PML body under heat shock conditions, if at some level misfolded, seems to be only correlative and does not mimic DRiPs behavior and dynamics.

Reply: To extend our observations from OP-puro labelled DRiPs to other misfolded proteins we used: 1) endogenous polyUb proteins (in cells exposed to heat shock without co-treatment with OP-puro; Figure 3B, C); 2) mCherry-VHL, a misfolded model substrate (Figure 6A-F); 3) NLuc-GFP (Appendix Figure S2). Upon stress, all these proteins showed subcellular distribution similar to the ones of polyUb proteins and DRiPs. Since these observations are correlative, we rephrased the text as follows: "During the recovery phase at normal temperature, NLuc-GFP was cleared from nucleoli and PML-NBs (Appendix Figure S2), similar to DRiPs and newly synthesized polyUb proteins".

The authors observed that the mobility of PML-GFP measured by FRAP was dramatically reduced by >50% in HeLa cells treated with proteasome inhibitor MG132 and OP-Puro in contrast to cells treated with the translation inhibitor cycloheximide. It has been reported that proteins localized in PML body move to the nucleolus upon MG132 treatment. Therefore, observed slowdown effect of PML-GFP mobility could be related to the reorganization of PML body structural integrity and might not be directly linked to the accumulation of DRiPs.

Reply: To monitor the presence of misfolded proteins inside PML-NBs and assess their direct impact on PML-GFP dynamics, we performed live-cell imaging in cells expressing mCherry-VHL, used as misfolded model substrate (Figure 6A). We now included in revised Figure 6A a picture showing that, in cells co-treated with cycloheximide, mCherry-VHL does not accumulate at PML-NBs (see revised Figure 6A); this correlates with FRAP analysis of PML-GFP mobility (compare Figure 6C and D) and strongly supports our interpretation that newly synthesized misfolded proteins that accumulate at PML-NBs affect their mobility.

The authors do not indicate which isoform(s) of PML protein they targeted for depletion by siRNA. Since there are many PML isoforms with different mobility, this might be an issue for FRAP comparison and overall mobility.

Reply: For our studies, we used a Dharmacon SMARTpool: ON-TARGETplus PML siRNA (L-006547-00-0005), which consists of four individual sequences targeting the following PML transcripts (<https://dharmacon.horizondiscovery.com/rnai/sirna/on-targetplus/on-targetplus-sirna-reagents-human/?sourceId=entrezgene/5371>):

NM_033244 Homo sapiens promyelocytic leukemia (PML), transcript variant 5, mRNA
NM_033247 Homo sapiens promyelocytic leukemia (PML), transcript variant 8, mRNA
NM_033249 Homo sapiens promyelocytic leukemia (PML), transcript variant 10, mRNA
NM_033250 Homo sapiens promyelocytic leukemia (PML), transcript variant 11, mRNA
NM_033246 Homo sapiens promyelocytic leukemia (PML), transcript variant 7, mRNA
NM_033240 Homo sapiens promyelocytic leukemia (PML), transcript variant 2, mRNA
NM_033238 Homo sapiens promyelocytic leukemia (PML), transcript variant 1, mRNA

NM_002675 Homo sapiens promyelocytic leukemia (PML), transcript variant 6, mRNA
NM_033239 Homo sapiens promyelocytic leukemia (PML), transcript variant 9, mRNA

Concerning protein detection by immunoblotting, we used the PML antibody (abcam 179466; observed band size: 50-110 kDa). This antibody was raised using a recombinant fragment that covers the region between amino acids 150-400 of the human PML protein, which is common to all PML isoforms (7). According to the manufacturer, the observed band sizes are from 50-110 kDa (https://www.abcam.com/PML-Protein-antibody-EPR16792-ab179466.html#description_images_1). We now show that PML siRNA treatment leads to a global reduction in the levels of PML (see revised Figure 4A).

VCP depletion significantly increases the number of PML bodies even in control cells.

Reply: VCP is required to degrade DRiPs (8). It is thus plausible that VCP depletion may lead to the accumulation of DRiPs at PML-NBs, whose number slightly increases already in resting cells as pointed-out by the referee.

Referee #3 (Report for Author)

Mediani and colleagues provide the characterization of defective ribosomal products - DRiPs - which are the major aggregation-prone proteins within cells. They report that DRiPs passively diffuse to the nucleus, accumulating to nucleoli and PML bodies, where they are processed by chaperons and proteasomal components. If this processing fails, accumulation of DRiPs promotes amyloidogenesis and liquid-solid transition of PML bodies, with the impairment of the all ubiquitin system, leading to impairment of ubiquitin-dependent events within the cells, including the DNA damage and repair processes.

Although the study is potentially of interest, the main conclusions are based on data obtained using a single experimental condition (OP-puro), which is quite harsh and artificial, leading to a general unbalance in protein homeostasis. Moreover, most of the results are based on a single technique, i.e. immunofluorescence, contributing to the weakness of this study. Many additional experiments, using different experimental approaches and conditions, should be done in support of the main findings.

Major points

1. Figure 1. The treatment with OP-puro, which allows to detect the fate of the DRiPs within the cell, blocks the synthesis of all cellular proteins and produces a huge amount of premature terminated proteins, which is highly artefactual. The authors should rather use a milder treatment of OP-puro, allowing the tagging of the misfolded proteins but in a more 'physio-pathological' context.

Reply: We performed a time-curve and a concentration-curve experiment. First, we incubated the cells with a high concentration of OP-puro (50 μ M, twice the concentration used throughout our studies), but for shorter times (from 1 min to 15 min): under these conditions, we could detect DRiPs inside nucleoli, visualized using fibrillarin, already after 5 min of treatment, with a progressive accumulation in time (see below).

DRiPs accumulate inside the nucleolus within few minutes (OP-puro 50 μ M)

DRiP labelling in HeLa cells that were treated with OP-puro (50 μ M) for 1, 5, 10 or 15 min. Nucleoli were visualized using a fibrillarin specific antibody. Note that, using this concentration of OP-puro, DRiPs can be detected inside nucleoli already after 5 min of treatment.

Second, as suggested by referee#3, we used lower concentrations of OP-puro. We confirmed the accumulation of DRiPs inside nucleoli in GFP-NCL HeLa Kyoto cells treated with a 5 times lower concentration of OP-puro (5 μ M instead of 25 μ M; see revised Figure 1B). Next, we could detect DRiPs inside nucleoli and at PML-NBs in cells exposed for 2 hrs to HS and treated with OP-puro 5 μ M and 10 μ M, respectively (see figure below).

Lowering OP-puro concentration does not affect DRiP accumulation at PML-NBs

HeLa cells were left untreated or treated for 2 hrs with HS and OP-puro at two different concentrations: 5 and 10 μ M, respectively. DRiPs accumulate at PML-NBs and inside nucleoli also when lower concentration of OP-puro are used.

We included in our revised manuscripts additional STED super-resolution microscopy data obtained in cells treated with OP-puro (5 μ M) alone or with heat shock or the proteasome inhibitor MG132. These new STED

data confirm the compartmentalization and colocalization of DRiPs with PML-NBs upon proteotoxic stress (see revised Figures 1B, 3A and E, lower panels).

Concerning the comment *“the main conclusions are based on data obtained using a single experimental condition (OP-puro), which is quite harsh and artificial”*, we observed nucleolar amyloidogenesis and accumulation of newly synthesized polyUb proteins in cells that were only subjected to heat shock (without addition of OP-puro; Figure 3B-D). We also included in our revised manuscript STED super-resolution microscopy data showing the accumulation of endogenous polyUb proteins at PML-NBs in cells exposed to heat shock alone (no OP-puro; Figure 3C). Moreover, we now show in cells exposed to heat shock alone that the endogenous polyUb proteins accumulating at PML-NBs convert them into an Amylo-Glo positive state (see revised Figure 5, new panel G). Here, we provide additional data for the referee showing that after 3 hrs of exposure to heat shock, both nucleoli and PML-NBs are positive to Amylo-Glo.

Amylo-Glo staining of PML-GFP HeLa Kyoto cells that were subjected to heat shock (HS) at 42°C for 3 hrs. Confocal images showing that Amylo-Glo and PML-GFP partly colocalize and formation of Amylo-Glo positive NoABs.

Combined, these data further support our interpretation that nucleoli and PML-NBs act as overflow compartments for DRiPs, excluding the hypothesis that such compartmentalization is an artefactual response to high amounts of prematurely terminated proteins.

Most of the DRiPs seems to go into the nucleus, although it would be expected to see a large part of them in the cytosol, as also suggested by the Western blot in Fig. 1B. It is not clear which is the portion and the size (molecular weight) of nucleolar DRiPs compared to the nuclear ones (not shown) and to the cytosolic fractions. Please, provide biochemical analysis of the distribution of DRiPs in different cell compartments: cytosol, nucleus, nucleoli.

Reply: The nucleoli isolation protocol initially used for our studies did not allow to separate nucleoplasmic proteins from cytoplasmic proteins (9). Using this protocol, we obtained a combined cytoplasmic and nucleoplasmic (CN) fraction and a nucleolar fraction. We therefore used another protocol that allowed us to obtain distinct cytosolic, nucleoplasmic and nucleolar fractions (10). As pointed out correctly by the referee, using this protocol we now show that the majority of DRiPs are cytosolic (see revised Figure 1C). Importantly, we confirm that the pool of DRiPs that accumulates in the nucleoplasm and nucleoli is characterized by a low molecular weight (see revised Figure 1C). These results further support our interpretation that low molecular weight DRiPs freely diffuse into the nucleus.

2. Figure 2-7. General comments for all the figures. Most of the experiments are done in different combinations of very harsh conditions (treatments with CHX, heat shock, MG132, OP-puro), which make difficult to correctly interpret the data.

Reply: As previously mentioned, we observed recruitment of endogenous newly synthesized polyUb proteins in the nucleus of cells that were only subjected to heat shock at 42°C for 2 hrs (without addition of OP-puro;

Figure 3B-D). Treatment of mammalian cells at 42°C for 1-3 hrs has been widely used to study how cells respond and adapt to heat shock and heat-shock induced protein misfolding (11-14). Concerning proteasome inhibition, we employed a relatively low concentration of MG132 (10 μM), also widely used throughout the scientific literature (15, 16). Finally, similar combinations of heat shock, puromycin and cycloheximide have been recently used to investigate how heat shock (42 and 43°C) affects transcription, with intriguing findings implicating active translation and nascent protein ubiquitination in transcription regulation (17).

Next, to further support our interpretation that misfolded proteins that accumulate inside nucleoli then convert into an amyloid-like state, we tested the effect of GFP-tagged poly-GR (GFP-GR50). GFP-GR50 is a dipeptide repeat (DPR) that accumulates inside the nucleolus and originates from the GGGGCC (G₄C₂) repeat expansion in a noncoding region of C9ORF72, which represents the most common cause of amyotrophic lateral sclerosis (ALS) and frontotemporal dementia (FTD) (18). Nucleoli that accumulate GFP-GR50, in absence of any other type of stress, convert into an amyloid-like state (see new panel I, revised Figure 2). By contrast, nucleoli in cells expressing GFP and nucleoli that accumulate the ribosomal protein RPL23a-GFP do not undergo amyloidogenesis (see revised Figure 2, new panel I). These data suggest that conversion of nucleoli into an amyloid-like state in cells expressing GFP-GR50 could contribute to toxicity together with changes in nucleolar functions and nucleocytoplasmic transport, as previously documented (18, 19).

3. Figure 3. Quantification of the co-localization data are missing and should be provided. DRiPs and PML do not really co-localize but are rather adjacent, which could be in principle an interesting observation. The ubiquitination of misfolded proteins has been largely described, therefore the co-localization of DRiPs and FK1 is not surprising.

Reply: The number of DRiPs foci, their enrichment for polyUb proteins (FK1), as well as the number of PML-NBs enriched for polyUb proteins have been quantified using a high-content automated imaging assay (ScanR, Olympus). High-content automated quantifications are reported in Figure 4B, C, Figure 5C, E and Fig. EV4D. In addition, we quantified the % of cells with FK1 nuclear foci upon stress; these quantifications are reported in Figure 3D, F and Figure 5B, D.

Why there is no FK1 signal in the nucleoli, which are full of DRiPs? How are the DRiPs degraded in the nucleoli, considering that both ubiquitin (FK1 staining) and the proteasome (20S) are excluded?

Reply: It has been previously published that proteasome-mediated degradation does not occur inside the nucleolus; instead, it was proposed to occur in discrete foci within the nucleoplasm, including PML-NBs (20-22). Our data are in line with these findings and suggest that misfolded proteins are extracted from the nucleolar subcompartments for disposal. In agreement with this model, we find that chaperones are recruited inside the nucleolus during stress, when amyloidogenesis occurs, and are required for the clearance of misfolded proteins from A-bodies during the recovery phase after stress (our manuscript) (3). Chaperones are also recruited at PML-NBs to clear DRiPs. We now show that during the recovery phase, inhibition of HSP70 ATPase activity leads to the accumulation of DRiPs and polyUb conjugates at PML-NBs. Of note, only the pool of DRiPs that is compartmentalized at PML-NBs, and not the one that persists inside nucleoli, colocalizes with polyUb conjugates (Appendix Figure S3E). Combined with the finding that 20S proteasomes concentrations are high in proximity to PML-NBs, these data strongly support our interpretation that degradation of polyUb proteins does not occurs inside nucleoli.

Please, provide biochemical analysis showing the ubiquitination of DRiPs.

It has been previously published that DRiPs are degraded by proteasomes (23). In addition, in a series of elegant experiments, Wang et al. recently demonstrated that polysome-associated puromycylated nascent polypeptides are ubiquitinated in cells (24). Using fluorescently labeled puromycin, the authors estimated that circa 15% of the total nascent polypeptides are co-translationally ubiquitinated. This percentage further increases upon stress conditions that induce protein misfolding (24). Finally, the authors found that puromycylated nascent chains mainly contain the K48-linked polyubiquitin chains, which is consistent with

the targeting of these products to the proteasome for degradation. Of note, the FK1 antibody that we use in our microscopy studies recognizes K48-linked polyubiquitinated and monoubiquitinated proteins, but not free ubiquitin.

To substantiate our microscopy data with biochemical data, we pulled-down DRiPs from the nucleoplasmic fraction and nucleoli using an anti-puromycin antibody. We compared non-treated cells versus cells treated with puromycin alone or with MG132. Revised Figure EV3, new panel B, shows that only nucleoplasmic DRiPs are ubiquitinated when cells are co-treated with puromycin and the proteasome inhibitor MG132. In addition, we show that nuclear puromycylated nascent chains are degraded by the proteasomes, which requires their polyubiquitination as targeting signal. By contrast, inhibition of lysosome acidification with ammonium chloride had no effect on nuclear DRiP clearance (see revised Figure EV3, new panel C). These results perfectly fit with our microscopy observations. These new biochemical data further support our conclusion that the pool of DRiPs that accumulates at PML-NBs is ubiquitinated.

4. Figure 5. Again, no data on direct ubiquitination of DRiPs.

Reply: See comment above (Figure EV3C).

5. Figure 6. FRAP technique measures the mobility of a specific protein over time. Using this as readout for the conversion of PML bodies from liquid to solid is a bit limited. Additional proofs should be provided.

Reply: First, we now include in our revised manuscript data showing that DRiPs that accumulate at PML-NBs convert into an amyloid-like state and can be detected using the Amylo-Glo dye (see revised Figure 5, new panel F). Importantly, preventing the accumulation of DRiPs at PML-NBs by co-incubating the cells with proteasome and translation inhibitors, also prevented amyloidogenesis at PML-NBs. We then show that, similar to NoABs (Figure 2), amyloidogenesis at PML-NBs is a reversible process. In fact, PML-NBs lost their positivity to Amylo-Glo during the recovery time after stress, when DRiPs were cleared, but not when DRiP clearance was impaired by inhibition of HSP70 (see revised Figure 5, new panel F). These data perfectly fit with the FRAP analysis of PML-GFP mobility shown in Figure 6.

Moreover, we show that Amylo-Glo positive foci overlapping with PML-NBs can also be observed in cells exposed to HS alone, without addition of OP-puro (see revised Figure 5, new panel G); this result further supports our interpretation that PML-NBs are stress-responsive overflow compartments that sequester misfolded proteins during stress conditions, excluding the hypothesis that this response only occurs due to huge amount of premature terminated proteins generated with OP-puro.

Second, an increase in the number of PML-NBs in response to stress, including heat shock, DNA damage and viral infection, has extensively documented (25). We confirmed these observations. Of note, during the recovery time after stress, liquid-like PML-NBs rapidly dissolved and returned to basal numbers. By contrast, when proteasome or chaperone function were inhibited during the recovery time, not only PML-NBs became solid (as measured by FRAP) and acquired amyloid-like properties, but their number did not decrease as efficiently as in control cells (Figure EV4E, F). Thus, the persistence in time of PML-NBs can be used as an additional method to evaluate their solidification. This type of analysis is widely used to study the conversion of other types of membraneless organelles, such as stress granules, into an aggregated-like state (5, 26, 27). We quantified the number of PML-NBs upon stress and during the recovery time after stress, using a high-content automated imaging assay (ScanR, Olympus). These quantifications are reported in Figure EV4E, F, as well as in Figure 5C, E. These results were also further validated by live-cell imaging studies in cells expressing PML-GFP (Movies EV1-5).

6. Figure 7. This figure recapitulates data already present in the literature. Blocking degradation of ubiquitinated proteins by using the proteasome inhibitor MG132 results in the depletion of the pool of free ubiquitin, thereby affecting all ubiquitin-based processes within the cells, including DNA damage response and DNA repair. Indeed, both H2A ubiquitination and recruitment of DDR factors to DNA lesions are largely impaired. See as representative references Mailand et al, Cell 2007; Chroma et al, Oncogene 2017.

Reply: We agree with the referee that previous groups published that the local recruitment of DDR factors to DNA damage sites is impaired by depletion of nuclear ubiquitin. However, the majority of these studies (included the ones reported by the referee) were performed in cells exposed to DNA damaging agents that cause DNA double-strand breaks such as e.g. ionizing irradiation. Here, we do not treat the cells with DNA damaging agents. In line with these findings, we demonstrate that also the recruitment of repair factors at spontaneous DNA lesions that form in growing cells depends on the availability of ubiquitin; importantly, we provide strong evidence for a competition between nuclear proteostasis and DNA damage repair. The novelty of our results stands in the finding that a fraction of DRiPs is continuously targeted to the nucleus; if not properly handled, DRiPs limit the available pool of ubiquitin and pose a major threat to cells, which then can no longer maintain a healthy genome.

Minor points

1. The Result section contains extensive discussion, which should be limited and moved to the Discussion section.

Reply: We moved from the result section to the discussion the following sentences:

“Hence, nuclear proteostasis and genome integrity are two competing processes that have to be well balanced to keep a cell in a healthy state.”

“When this process fails, PML-NBs adopt solid-like properties and immobilize chaperones, ubiquitin and 20S proteasomes, potentially compromising their nuclear functions.”

2. In some parts of the Results, the reference to Figures is a bit confusing.

Reply: We ameliorated the reference to Figures with textual revision.

REFERENCES

1. Jacob MD, Audas TE, Uniacke J, Trinkle-Mulcahy L, Lee S. Environmental cues induce a long noncoding RNA-dependent remodeling of the nucleolus. *Mol Biol Cell*. 2013;24(18):2943-53.
2. Audas TE, Jacob MD, Lee S. Immobilization of proteins in the nucleolus by ribosomal intergenic spacer noncoding RNA. *Mol Cell*. 2012;45(2):147-57.
3. Audas TE, Audas DE, Jacob MD, Ho JJ, Khacho M, Wang M, et al. Adaptation to Stressors by Systemic Protein Amyloidogenesis. *Dev Cell*. 2016;39(2):155-68.
4. Latonen L, Moore HM, Bai B, Jaamaa S, Laiho M. Proteasome inhibitors induce nucleolar aggregation of proteasome target proteins and polyadenylated RNA by altering ubiquitin availability. *Oncogene*. 2011;30(7):790-805.
5. Ganassi M, Mateju D, Bigi I, Mediani L, Poser I, Lee HO, et al. A Surveillance Function of the HSPB8-BAG3-HSP70 Chaperone Complex Ensures Stress Granule Integrity and Dynamism. *Mol Cell*. 2016;63(5):796-810.
6. Stavreva DA, Kawasaki M, Dundr M, Koberna K, Muller WG, Tsujimura-Takahashi T, et al. Potential roles for ubiquitin and the proteasome during ribosome biogenesis. *Mol Cell Biol*. 2006;26(13):5131-45.
7. Nisole S, Maroui MA, Mascle XH, Aubry M, Chelbi-Alix MK. Differential Roles of PML Isoforms. *Front Oncol*. 2013;3:125.
8. Verma R, Oania RS, Kolawa NJ, Deshaies RJ. Cdc48/p97 promotes degradation of aberrant nascent polypeptides bound to the ribosome. *eLife*. 2013;2:e00308.
9. Li ZF, Lam YW. A new rapid method for isolating nucleoli. *Methods Mol Biol*. 2015;1228:35-42.
10. Andersen JS, Lyon CE, Fox AH, Leung AK, Lam YW, Steen H, et al. Directed proteomic analysis of the human nucleolus. *Curr Biol*. 2002;12(1):1-11.
11. Kose S, Furuta M, Imamoto N. Hikeshi, a nuclear import carrier for Hsp70s, protects cells from heat shock-induced nuclear damage. *Cell*. 2012;149(3):578-89.
12. Rizvi SM, Mancino L, Thammavongsa V, Cantley RL, Raghavan M. A polypeptide binding conformation of calreticulin is induced by heat shock, calcium depletion, or by deletion of the C-terminal acidic region. *Mol Cell*. 2004;15(6):913-23.
13. Sarge KD, Murphy SP, Morimoto RI. Activation of heat shock gene transcription by heat shock factor 1 involves oligomerization, acquisition of DNA-binding activity, and nuclear localization and can occur in the absence of stress. *Mol Cell Biol*. 1993;13(3):1392-407.
14. Welch WJ, Suhan JP. Morphological study of the mammalian stress response: characterization of changes in cytoplasmic organelles, cytoskeleton, and nucleoli, and appearance of intranuclear actin filaments in rat fibroblasts after heat-shock treatment. *J Cell Biol*. 1985;101(4):1198-211.
15. Jena KK, Kolapalli SP, Mehto S, Nath P, Das B, Sahoo PK, et al. TRIM16 controls assembly and degradation of protein aggregates by modulating the p62-NRF2 axis and autophagy. *EMBO J*. 2018;37(18).
16. Watanabe Y, Tsujimura A, Taguchi K, Tanaka M. HSF1 stress response pathway regulates autophagy receptor SQSTM1/p62-associated proteostasis. *Autophagy*. 2017;13(1):133-48.

17. Aprile-Garcia F, Tomar P, Hummel B, Khavaran A, Sawarkar R. Nascent-protein ubiquitination is required for heat shock-induced gene downregulation in human cells. *Nat Struct Mol Biol.* 2019;26(2):137-46.
18. Freibaum BD, Lu Y, Lopez-Gonzalez R, Kim NC, Almeida S, Lee KH, et al. GGGGCC repeat expansion in C9orf72 compromises nucleocytoplasmic transport. *Nature.* 2015;525(7567):129-33.
19. Kwon I, Xiang S, Kato M, Wu L, Theodoropoulos P, Wang T, et al. Poly-dipeptides encoded by the C9orf72 repeats bind nucleoli, impede RNA biogenesis, and kill cells. *Science.* 2014;345(6201):1139-45.
20. Scharf A, Rockel TD, von Mikecz A. Localization of proteasomes and proteasomal proteolysis in the mammalian interphase cell nucleus by systematic application of immunocytochemistry. *Histochem Cell Biol.* 2007;127(6):591-601.
21. Rockel TD, Stuhlmann D, von Mikecz A. Proteasomes degrade proteins in focal subdomains of the human cell nucleus. *J Cell Sci.* 2005;118(Pt 22):5231-42.
22. Guo L, Giasson BI, Glavis-Bloom A, Brewer MD, Shorter J, Gitler AD, et al. A cellular system that degrades misfolded proteins and protects against neurodegeneration. *Mol Cell.* 2014;55(1):15-30.
23. Schubert U, Anton LC, Gibbs J, Norbury CC, Yewdell JW, Bennink JR. Rapid degradation of a large fraction of newly synthesized proteins by proteasomes. *Nature.* 2000;404(6779):770-4.
24. Wang F, Durfee LA, Huibregtse JM. A cotranslational ubiquitination pathway for quality control of misfolded proteins. *Mol Cell.* 2013;50(3):368-78.
25. Lallemand-Breitenbach V, de The H. PML nuclear bodies: from architecture to function. *Curr Opin Cell Biol.* 2018;52:154-61.
26. Mackenzie IR, Nicholson AM, Sarkar M, Messing J, Purice MD, Pottier C, et al. TIA1 Mutations in Amyotrophic Lateral Sclerosis and Frontotemporal Dementia Promote Phase Separation and Alter Stress Granule Dynamics. *Neuron.* 2017;95(4):808-16 e9.
27. Lee KH, Zhang P, Kim HJ, Mitrea DM, Sarkar M, Freibaum BD, et al. C9orf72 Dipeptide Repeats Impair the Assembly, Dynamics, and Function of Membrane-Less Organelles. *Cell.* 2016;167(3):774-88 e17.

Thank you for submitting your revised manuscript for our consideration, and apologies for the delay in its re-evaluation. It has now been seen once more by the original reviewers, whose comments are copied below. As you will see, all referees appreciate your responses and experimental additions as major improvements to the study, in light of which we shall be happy to offer eventual publication in The EMBO Journal. Nevertheless, referee 3 still retains significant reservations with regard to the data in Figure 7 and the major conclusions drawn from it, which -having discussed these issues with the other referees- I feel are well-taken and justify addressing in a further round of minor revision. The two key points here are that the papers mentioned by the referee (Mailand et al, Chroma et al) need to be cited and discussed as conceptual precedent, even if the data on MG132 effects in non-damaged cells are only a minor aspect of these works; and more importantly, that Fig. 7 does indeed not seem to support major additive or single-agent effects of OP-Puromycin, as compared to treatment with proteasome inhibitor itself. While I do not agree with referee 3 that Figure 7 should be removed altogether, I do feel that this issue warrants reconsideration of the interpretations from these experiments and altering the conclusions in the abstract and results section, as well as more cautious discussion throughout the text.

Therefore, please answer once more with a point-by-point response letter to the remaining referee concerns, and incorporate the appropriate changes (using the "Track Changes" option) in the attached Word document of the text.

REFeree REPORTS

Referee #1:

affect cell function. They show that many DRiPs, in particular short DRiPs, freely diffuse into the nucleus and partition into the nucleolus. There, they lead to protein aggregation and the formation of nucleolar aggresomes or amyloid-bodies, which the authors here show to be the same body and hence call NoABs. The combination of DRiPs with heat or proteotoxic stress results in the accumulation of DRiPs in PML bodies, which transition from a liquid into a solid state, which is characterized by immobilization by PML-associated proteins, but also ubiquitin, proteasomes and other PQC factors. These factors are essential for marking and protection of DNA lesions, and the authors demonstrate that proteostasis processes and DNA repair compete for these factors under stress conditions.

These findings show an interesting mechanistic link between nuclear proteostasis and defective DNA repair compartments. This link occurs via liquid membraneless compartments that change their material properties upon stress via the incorporation of the DRiPs. This manuscript fits into a series of publications that showed a link between stress conditions, aberrant phase transitions and changes in material properties of liquid organelles. Importantly, not only stress granules are affected as also shown in a series of manuscripts on C9orf72 repeat expansions.

This is an important manuscript that produces beautiful data in an elegant series of experiments. In my opinion, it is suitable for publication.

Referee #2:

In my opinion, the revised version of the manuscript is suitable for publication in EMBO J.

Referee #3:

This reviewer appreciates the new experiments (time-curve and dose-curve), as well as the use of STED microscopy, which consolidate the data and improve the manuscript.

Regarding the general comment on the use of harsh conditions in most of the experiments, this reviewer still considers it an issue. The manuscript is largely based on combinations of multiple cell

treatments, thereby generating data difficult to interpret. As an example, they used in many experiments different combination of drugs (HS, OP-Puro, CHX, ActD) in the presence of proteasome inhibitor MG132 (10 uM, 4 hr; Fig. 3E, Fig. 4B, 4C, 4F, 4G, Fig. 5A-C, 5F, Fig. 6A, 6C-H, Fig. 7A-E), which per se strongly affects the ubiquitin - and cell - homeostasis.

The major concerns are on the results presented in Fig. 7 and on the conclusions drawn based on them. The data shown in Fig. 7 (and also Fig. 3E) can be explained by the sole treatment with the proteasome inhibitor MG132 (10 uM, 4 hr), which they used, in combinations with other treatments, in all the experiments performed (Fig. 7A-E), importantly also in the colony formation assay. As clear proof of this, treatment with OP-Puro alone did not exert any effect (Fig. 7B), being very similar to control. Thus, the data shown are likely due to the effect of proteasome inhibition and the consequent accumulation of ubiquitinated proteins within the cells. In line with this, the Western blot in Fig. EV5C shows high molecular weight proteins rather than low molecular weight, as expected if they were mainly due to Puromycin treatment (Fig. 1 C). Thus, the accumulation of ubiquitinated conjugates leads to depletion of free ubiquitin, which is not available anymore for normal physiological processes, including taking care of replication problems or DNA damage normally occurring in cycling cells.

Moreover, as already mentioned, this is not an original observation, since it was already reported in the literature by different publications, including Mailand et al, *Cell* 2007 and Chroma et al, *Oncogene* 2017. Although the main message of the two cited papers is related to conditions of DNA damage, the authors also performed experiments in untreated conditions (i.e no genotoxic stress). In Mailand et al, the authors demonstrated that treatment with MG132 (5 uM, for 0, 30, 60, 90, 120 min) inhibits histone ubiquitination (H2A, H2AX, H2AZ and H2B). In Chroma et al, the authors reported that MG132 (5 uM, 2 hr) abolished 53BP1 foci formation in undamaged cells (no genotoxic stress) in 3 different cell lines (MDA-MB-231, U2OS, BJ, Fig. 1b, Suppl. Fig. S1D, S3).

In order to avoid misleading message to the readers, it is recommendable to remove the data presented in Fig. 7 and the conclusions based on those data from the manuscript (Summary, Results), and to discuss the possible implications of their findings, for ubiquitin turnover and for the maintenance of genome stability, in the Discussion.

Manuscript EMBOJ-2018-101341R: “Defective ribosomal products (DRiPs) challenge nuclear function by impairing nuclear condensate dynamics and immobilizing ubiquitin” by Mediani et al.**Point-by-point reply to referees' comments:****Referee #1:**

They show that many DRiPs, in particular short DRiPs, freely diffuse into the nucleus and partition into the nucleolus. There, they lead to protein aggregation and the formation of nucleolar aggresomes or amyloid-bodies, which the authors here show to be the same body and hence call NoABs. The combination of DRiPs with heat or proteotoxic stress results in the accumulation of DRiPs in PML bodies, which transition from a liquid into a solid state, which is characterized by immobilization by PML-associated proteins, but also ubiquitin, proteasomes and other PQC factors. These factors are essential for marking and protection of DNA lesions, and the authors demonstrate that proteostasis processes and DNA repair compete for these factors under stress conditions.

These findings show an interesting mechanistic link between nuclear proteostasis and defective DNA repair compartments. This link occurs via liquid membraneless compartments that change their material properties upon stress via the incorporation of the DRiPs. This manuscript fits into a series of publications that showed a link between stress conditions, aberrant phase transitions and changes in material properties of liquid organelles. Importantly, not only stress granules are affected as also shown in a series of manuscripts on C9orf72 repeat expansions.

This is an important manuscript that produces beautiful data in an elegant series of experiments. In my opinion, it is suitable for publication.

Reply: We are pleased to hear that referee #1 likes our manuscript and recommends publication.

Referee #2:

In my opinion, the revised version of the manuscript is suitable for publication in EMBO J.

Reply: We are pleased to hear that referee #2 recommends publication of our revised manuscript.

Referee #3:

This reviewer appreciates the new experiments (time-curve and dose-curve), as well as the use of STED microscopy, which consolidate the data and improve the manuscript.

Regarding the general comment on the use of harsh conditions in most of the experiments, this reviewer still considers it an issue. The manuscript is largely based on combinations of multiple cell treatments, thereby generating data difficult to interpret. As an example, they used in many experiments different combination of drugs (HS, OP-Puro, CHX, ActD) in the presence of proteasome inhibitor MG132 (10 μ M, 4 hr; Fig. 3E, Fig. 4B, 4C, 4F, 4G, Fig. 5A-C, 5F, Fig. 6A, 6C-H, Fig. 7A-E), which per se strongly affects the ubiquitin - and cell - homeostasis.

Reply: We report nucleolar amyloidogenesis and compartmentalization of endogenous polyubiquitinated proteins at PML-NBs also in cells exposed to heat shock alone, and not treated with OP-puro or exposed to combination of drugs (Figure 2A, G, Figure 3B, C, Figure 5G).

The major concerns are on the results presented in Fig. 7 and on the conclusions drawn based on them. The data shown in Fig. 7 (and also Fig. 3E) can be explained by the sole treatment with the proteasome inhibitor MG132 (10 uM, 4 hr), which they used, in combinations with other treatments, in all the experiments performed (Fig. 7A-E), importantly also in the colony formation assay. As clear proof of this, treatment with OP-Puro alone did not exert any effect (Fig. 7B), being very similar to control. Thus, the data shown are likely due to the effect of proteasome inhibition and the consequent accumulation of ubiquitinated proteins within the cells. In line with this, the Western blot in Fig. EV5C shows high molecular weight proteins rather than low molecular weight, as expected if they were mainly due to Puromycin treatment (Fig. 1 C). Thus, the accumulation of ubiquitinated conjugates leads to depletion of free ubiquitin, which is not available anymore for normal physiological processes, including taking care of replication problems or DNA damage normally occurring in cycling cells.

Moreover, as already mentioned, this is not an original observation, since it was already reported in the literature by different publications, including Mailand et al, Cell 2007 and Chroma et al, Oncogene 2017. Although the main message of the two cited papers is related to conditions of DNA damage, the authors also performed experiments in untreated conditions (i.e no genotoxic stress). In Mailand et al, the authors demonstrated that treatment with MG132 (5 uM, for 0, 30, 60, 90, 120 min) inhibits histone ubiquitination (H2A, H2AX, H2AZ and H2B). In Chroma et al, the authors reported that MG132 (5 uM, 2 hr) abolished 53BP1 foci formation in undamaged cells (no genotoxic stress) in 3 different cell lines (MDA-MB-231, U2OS, BJ, Fig. 1b, Suppl. Fig. S1D, S3).

In order to avoid misleading message to the readers, it is recommendable to remove the data presented in Fig. 7 and the conclusions based on those data from the manuscript (Summary, Results), and to discuss the possible implications of their findings, for ubiquitin turnover and for the maintenance of genome stability, in the Discussion.

Reply: As correctly pointed out by the referee, Chroma et al. (2016) showed in Figures 1b, S1d and S3 that treatment of the MDA-MB-231, U2OS and BJ cell lines with MG132, in absence of genotoxic stress, decreased the number of 53BP1 foci. Moreover, Mailand et al. (2007; Figure S6B) published that treatment of U2OS cells with MG132 decreases the levels of H2A-Ub. Thus, these authors demonstrated in different cell lines that ubiquitin starvation due to proteotoxic stress attenuates H2A-Ub levels and 53BP1 recruitment to DNA damage sites. In our revised manuscript, we now clearly state that these findings were previously published by other researchers: “As previously shown by independent groups, we also found that the number of these 53BP1 foci strongly decreased upon proteasome inhibition with MG132, alone or combined with OP-puro (Fig 7A, B); and this, in turn, correlated with a reduction of H2A-Ub levels and an accumulation of polyUb substrates (Fig EV5C) (Chroma, Mistrik et al., 2016, Jacquemont & Taniguchi, 2007, Mailand, Bekker-Jensen et al., 2007, Mimnaugh, Chen et al., 1997)”.

However, Chroma et al. (2016) and Mailand et al. (2007) did not address the question whether, upon proteasome inhibition, the decreased H2A ubiquitination and the decreased number of 53BP1 foci are a consequence of impaired degradation of ubiquitinated pre-existing proteins or newly synthesized proteins, including DRiPs. In our manuscript we specifically addressed the role of newly synthesized proteins and DRiPs in decreasing the pool of free ubiquitin, at the expense of the DNA damage response, by co-treating the cells with the proteasome inhibitor MG132 and the translation inhibitor cycloheximide. Figure EV5D shows that inhibition of translation with cycloheximide rescues 53BP1 foci formation upon proteasome inhibition. In addition, Figure EV5C shows that co-treatment of the cells with cycloheximide

and MG132 partly restores H2A-Ub levels. Thus, the novelty of our findings stands in the demonstration that, if not cleared by the protein quality control, ubiquitinated newly synthesized proteins and ubiquitinated nucleoplasmic DRiPs limit the pool of free ubiquitin, potentially threatening genome stability.

Concerning the comment “*As clear proof of this, treatment with OP-Puro alone did not exert any effect (Fig. 7B), being very similar to control*”, our data show that in cells treated with OP-puro alone, DRiPs are transiently compartmentalized in nucleolar subcompartments; DRiPs are rapidly cleared by proteasomes from these compartments, without negative effects on nucleolar functionality (Figure 1A, B and Figure EV1D-F). Of note, in normally growing cells that are treated with OP-puro alone, we did not observe accumulation of ubiquitin inside the nucleoli. Under these conditions, ubiquitin is not depleted and can be recycled for other functions, including targeting 53BP1 to DNA damage sites. By contrast, when the clearance of DRiPs is impaired due to chaperone or proteasome inhibition, polyubiquitinated DRiPs are sequestered with 20S proteasomes at PML-NBs (Figure 5 and Appendix Figure S3E), resulting in free ubiquitin depletion. This aspect has been emphasized in the revised manuscript by textual revision: “Importantly, while upon MG132 treatment, polyubiquitinated DRiPs were sequestered with 20S at PML-NBs, upon treatment of the cells with OP-puro alone we did not observe accumulation of ubiquitin inside nucleoli and the pool of DRiPs that was transiently compartmentalized in nucleolar subcompartments was rapidly cleared by proteasomes. In agreement, treatment of the cells with OP-puro alone did not significantly affect 53BP1 foci formation (Fig 7B).”

Finally, in order to avoid misleading information, we rephrased our conclusions in the revised manuscript and we included citation to Chroma et al. (2016) and Mailand et al. (2007): “Upon proteasome inhibition, a link between decreased levels of free ubiquitin and defective formation of DNA repair compartments at fragile chromosomal sites was previously reported by independent groups (Chroma, Mistrik et al., 2016, Mailand, Bekker-Jensen et al., 2007). Our data identify newly synthesized proteins, including DRiPs, as the main species that upon proteasome inhibition deplete free ubiquitin, thereby compromising DNA damage sensing/repair.”

References

Chroma K, Mistrik M, Moudry P, Gursky J, Liptay M, Strauss R, Skrott Z, Vrtel R, Bartkova J, Kramara J, Bartek J (2016) Tumors overexpressing RNF168 show altered DNA repair and responses to genotoxic treatments, genomic instability and resistance to proteotoxic stress. *Oncogene* 36: 2405

Mailand N, Bekker-Jensen S, Faustrup H, Melander F, Bartek J, Lukas C, Lukas J (2007) RNF8 ubiquitylates histones at DNA double-strand breaks and promotes assembly of repair proteins. *Cell* 131: 887-900

Accepted

7th June 2019

Thank you for submitting your final revised manuscript for our consideration. I am pleased to inform you that we have now accepted it for publication in The EMBO Journal.

Corresponding Author Name: Serena Carra

Manuscript Number: EMBOJ-2018-101341